# DON'T "OVERTHINK" POINTWISE RERANKING: IS REASONING TRULY NECESSARY?

## ABSTRACT

With the growing success of reasoning models across complex natural language tasks, researchers in the Information Retrieval (IR) community have begun exploring how similar reasoning capabilities can be integrated into passage rerankers built on Large Language Models (LLMs). These methods typically employ an LLM to produce an explicit, step-by-step reasoning process before arriving at a final relevance prediction. But, *does reasoning actually improve pointwise reranking accuracy?* In this paper, we dive deeper into this question, studying the impact of the reasoning process by comparing reasoning-based pointwise rerankers (Rank1) to standard, non-reasoning pointwise rerankers (StandardRanker) under identical training conditions, and observe that StandardRanker generally outperforms Rank1. Building on this observation, we then study the importance of reasoning to Rank1 by disabling its reasoning process (Rank1-NoReason), and find that Rank1-NoReason is surprisingly more effective than Rank1. Examining the cause of this result, our findings reveal that pointwise reasoning rerankers are bottlenecked by the LLM's reasoning process, which pushes it toward polarized relevance scores and thus fails to consider the *relative* relevance of passages, a key factor for the accuracy of pointwise rerankers. The source code is in the supplementary materials and will be made public upon acceptance.

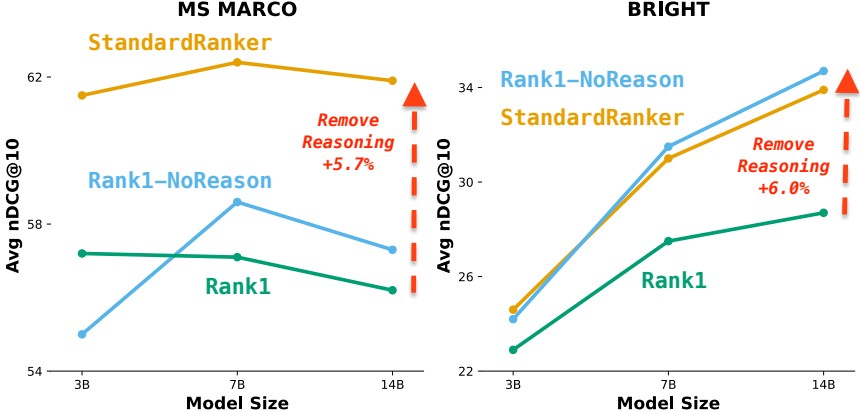

Figure 1: Average nDCG@10 of reasoning pointwise rerankers (Rank1) compared to their non-reasoning variants (StandardRanker and Rank1-NoReason) on MS MARCO and BRIGHT.

## 1 INTRODUCTION

Recently there has been a surge of interest in reasoning models such as DeepSeek-R1 (Guo et al., 2025), OpenAI's o3, Qwen3 (Yang et al., 2025a), and others. Through the generation of an explicit reasoning process—i.e., a chain-of-thought (CoT)—prior to producing its final response, reasoning models have shown strong performance across a wide range of complex natural language tasks such as mathematics (Yang et al., 2024).

Following this success, one area of focus for researchers in the information retrieval (IR) community has been on exploring how to best augment passage rerankers built on large language models (LLMs)

with reasoning capabilities (Weller et al., 2025; Zhuang et al., 2025), especially after the introduction of reasoning-intensive retrieval benchmarks such as BRIGHT (Su et al., 2025). At a high-level, given an input query and a passage (or list of passages), these *reasoning rerankers* are trained to first generate an explicit reasoning process before producing a final relevance score for the query-passage pair (or, if the input is a list of passages, an ordering of such list).

While various reasoning rerankers have been proposed and report strong gains, many of these works lack a thorough comparison to the *equivalent*—same backbone model, same training data—non-reasoning variants. Instead, the primary non-reasoning baselines are limited to passage rerankers built on previous generation LLMs — for example, RankLLaMA (Ma et al., 2024) (based on Llama-2), RankZephyr (Pradeep et al., 2023) (based on Zephyr), or MonoT5/RankT5 (Nogueira et al., 2020; Zhuang et al., 2023) (based on T5)—while the proposed models are leveraging the latest LLMs. Comparisons with equivalent, non-reasoning variants are typically relegated to ablation studies or are only studied for a limited subset of presented datasets, raising the question if reasoning rerankers are *truly* more effective or if the reported gains are simply due to improvements from using stronger backbone LLMs.

In this paper, we address this gap by revisiting baselines for Rank1 (Weller et al., 2025), a pointwise reasoning reranker that first generates a reasoning chain prior to producing a relevance score for an independent query-passage pair. We ask: *under equivalent training conditions (same backbone model, same training data), does the generation of a reasoning chain prior to making a relevance prediction actually improve pointwise reranking accuracy?* Our focus on Rank1 is simple: as it was the pioneering work in the reasoning reranking space, it has become a go-to baseline in follow-up research. Understanding its effectiveness in comparison to equivalent, non-reasoning counterparts is critical to ensure progress in developing more effective reasoning rerankers.[1]

To answer our research question, we compare Rank1 against two simple, yet effective non-reasoning reranker baselines which leverage the same backbone model and are trained with the same data: (1) **StandardRanker**, a standard LLM-based pointwise reranker that directly classifies query-passage pairs as relevant or irrelevant (Nogueira et al., 2020; Ma et al., 2024); and (2) **Rank1-NoReason**, a modified version of Rank1, in which the explicit reasoning process is disabled at inference time, effectively transforming Rank1 into a standard pointwise reranker. The central findings of our experiments can be summarized as follows and are shown in Figure 1:

- Under identical training setups, we find no general advantage of the reasoning process for pointwise reranking. Across multiple LLM sizes, StandardRanker was more effective in-domain and out-of-domain versus Rank1, on average.

- In fact, we find that reasoning can even degrade effectiveness for rerankers explicitly trained to reason (i.e., Rank1). As shown in Figure 1, Rank1-NoReason outperforms Rank1 in-domain (MS MARCO) for the 7B and 14B model sizes, and is always more effective out-of-domain (BRIGHT).

- Further investigation suggests that this likely stems from the reasoning process forcing the model towards *polarized* relevance scores which do not account for the *relative* relevance of passages. Our results show that while Rank1 is a better relevance classifier than Rank1-NoReason, Rank1-NoReason placed more emphasis on partial relevance scores, contributing to its better reranking accuracy.

Our findings build upon recent research in the NLP community which question the necessity of the reasoning process for LLMs (Ma et al., 2025; Sprague et al., 2025). We hope our work not only encourages future research that can improve reasoning for reranking tasks, but also highlights the importance of comparing against strong, simple, and fair baselines when developing new methodologies.

## 2 BACKGROUND: POINTWISE RERANKING WITH LLMS

In this section we provide brief background on pointwise rerankers (StandardRanker) and reasoning pointwise rerankers (Rank1), as they form the basis of our study.

---

[1]The authors do perform this comparison, but only for the 7B model, leaving the impact of model size unclear; they do not report results on individual datasets, so it's unclear which tasks benefit from reasoning; and they mention only a single low score of 17.5 on BRIGHT (page 7) without training details. This motivated us to run a more thorough baseline study.

**Preliminaries.** The goal of information retrieval (IR) is to identify relevant passages from a large collection of $n$ texts, denoted by $\mathcal{C} = \{P_1, P_2, \ldots, P_n\}$, given a user-issued query, $q$. Current IR systems typically employ a multi-stage pipeline where a first-stage retriever fetches an initial set of $k$ passages from $\mathcal{C}$ and a *reranker* reorders the top-$k$ passages $\{P_1, P_2, \ldots, P_k\}$, where $k \ll n$, to produce a more accurate ranking.

**StandardRanker.** StandardRanker is trained as a pointwise reranker, independently producing a relevance score, $R$, for a given query-passage pair. To train StandardRanker, the simplest approach is to directly fine-tune an LLM to produce the tokens "true" or "false" given a dataset of (query, passage, relevance label) triples, where "true" and "false" denote relevant or not relevant, respectively.

At inference, for each query-passage pair $(q, P_i)$ in the top-$k$, the probability of relevance, $R$, is computed by applying a softmax exclusively to the logits corresponding to the tokens "true" and "false":

$$R = \mathrm{softmax}\big(z_{\text{true}}(q, P_i), z_{\text{false}}(q, P_i)\big)_{\text{true}} \tag{1}$$

Here, $z_{\text{true}}(q, P_i)$ and $z_{\text{false}}(q, P_i)$ denote the logits assigned by the LLM for the "true" and "false" tokens, given input $(q, P_i)$. The subscript "true" after the softmax normalization indicates that only the probability assigned to the token "true" is considered for $R$. The passages are then sorted in descending order of $R$. We note that recent methods, such as RankLLaMA (Ma et al., 2024), train pointwise rerankers using hard negatives sampled from the top-ranking results of a first-stage retriever. However, as our goal is to keep the training setup identical to that of Rank1, which we describe next, we do not consider hard negatives.

**Rank1.** Rank1 builds upon the setup described for StandardRanker by fine-tuning an LLM to first generate a reasoning process, $r$, before producing the tokens "true" or "false". To do so, Rank1 is fine-tuned with a dataset of (query, passage, $r$, relevance label) quadruples.

Following Equation 1, $R$ is again computed by considering the softmax over the logits of the "true" and "false" tokens, but in this case, $R$ also considers the LLM's generated reasoning process, $r$:

$$R = \mathrm{softmax}\big(z_{\text{true}}(q, P_i, r_i), z_{\text{false}}(q, P_i, r_i)\big)_{\text{true}} \tag{2}$$

where $r_i$ is the reasoning process generated for input $(q, P_i)$. The passages are then reordered as described for StandardRanker.

## 3 DOES REASONING IMPROVE POINTWISE RERANKERS?

In this section we study the impact of reasoning on pointwise rerankers through two different lenses: (1) how does StandardRanker compare to Rank1 when trained under the same settings? And, (2) how is Rank1's reranking accuracy affected if we disable its reasoning process (Rank1-NoReason)? Through these two perspectives, we hope to shed light on different ways reasoning may influence pointwise reranking accuracy.

### 3.1 RQ1: STANDARDRANKER VS. RANK1

Our first experiment aims to understand the importance of reasoning from the training perspective. Specifically, if we train StandardRanker on the exact same data as Rank1, but omit the reasoning chain, how does performance compare? To answer this research question, we evaluate pointwise rerankers of varying sizes, with and without reasoning chains.

**Experiment Setup.** To train the pointwise rerankers, we leverage the training data provided by Weller et al. (2025).[2] The dataset augments MS MARCO (Bajaj et al., 2016) with reasoning chains generated by Deepseek R1 (Guo et al., 2025), which include R1's final relevance predictions. The dataset consists of approximately 386K quadruples in the following format: (query, passage, R1's reasoning chain, relevance label).

---

[2]https://huggingface.co/datasets/jhu-clsp/rank1-training-data

| | MS MARCO v1 | | MS MARCO v2 | | | Avg. |
|---|---|---|---|---|---|---|
| | DL19 | DL20 | DL21 | DL22 | DL23 | |
| BM25 | 50.6 | 48.0 | 44.6 | 26.9 | 26.3 | 39.3 |
| + Qwen2.5-3B | | | | | | |
| StandardRanker | **72.5** | **68.9**$^\dagger$ | **69.4**$^\dagger$ | **51.4**$^\dagger$ | **45.5**$^\dagger$ | **61.5** |
| Rank1 (our impl.) | 70.4 | 66.4 | 65.9 | 45.2 | 41.3 | 57.8 |
| Rank1 | 70.1 | 64.5 | 65.3 | 45.2 | 40.7 | 57.2 |
| + Qwen2.5-7B | | | | | | |
| StandardRanker | **74.6**$^\dagger$ | **70.0**$^\dagger$ | **70.9**$^\dagger$ | **50.3** | **46.3**$^\dagger$ | **62.4** |
| Rank1 (our impl.) | 70.3 | 64.3 | 65.9 | 45.6 | 41.1 | 57.4 |
| Rank1 | 68.4 | 65.1 | 65.8 | 44.7 | 41.6 | 57.1 |
| + Qwen2.5-14B | | | | | | |
| StandardRanker | **73.3**$^\dagger$ | **68.7**$^\dagger$ | **70.7**$^\dagger$ | **49.6**$^\dagger$ | **47.3**$^\dagger$ | **61.9** |
| Rank1 | 66.3 | 65.0 | 63.1 | 44.8 | 41.7 | 56.2 |

Table 1: In-domain performance of StandardRanker versus Rank1. Each Qwen2.5 model reranks the top-100 passages from BM25. **Bold** denotes best reranker under each Qwen2.5 model. $^\dagger$ denotes a statistically significant improvement over Rank1 based on a paired students t-test ($p < 0.05$) (Bassani, 2022).

For the backbone LLM, we leverage the Qwen2.5 base models (Yang et al., 2024) ranging from 3B to 14B model sizes. To train StandardRanker, we fine-tune using LoRA (Hu et al., 2022) for one epoch with rank 32 and alpha 64, using only the (query, passage, relevance label) triples, omitting R1's reasoning chain.[3] To ensure that any differences in Rank1's effectiveness are not due to differences in how models were trained, we additionally train our own variant of Rank1, Rank1 (our impl.), for the 3B and 7B model sizes following the same training process as StandardRanker, except now utilizing the full (query, passage, R1's reasoning chain, relevance label) quadruples.

We evaluate StandardRanker and Rank1 on in-domain and out-of-domain retrieval datasets. For in-domain evaluation, we leverage passage ranking datasets based on MS MARCO v1—TREC DL19 and TREC DL20 (Craswell et al., 2020; 2021b)—and based on MS MARCO v2—TREC DL21, TREC DL22, and TREC DL23 (Craswell et al., 2021a; 2022; 2023). For out-of-domain evaluation, we focus on BRIGHT (Su et al., 2025), a reasoning-intensive retrieval benchmark. We report nDCG@10, the official metric for both the MS MARCO and BRIGHT datasets.

At inference, models rerank the top-100 passages retrieved by BM25. For BRIGHT, models rerank passages retrieved by BM25 using queries expanded with GPT-4 CoT; however, following Weller et al. (2025), the rerankers are *not* provided the GPT-4 CoT. For MS MARCO, we implement BM25 using Pyserini (Lin et al., 2021) and for BRIGHT, we follow the implementation from the BRIGHT codebase. LLM training was performed using HuggingFace (Wolf et al., 2019) and inference with vLLM (Kwon et al., 2023).

**Results.** In Tables 1 and 2 we present the evaluation results for both in-domain and out-of-domain retrieval tasks. First, we note that Rank1 is comparable to Rank1 (our impl.), achieving a similar average nDCG@10 across MS MARCO and BRIGHT for the 3B and 7B model sizes. This validates that any differences in effectiveness between StandardRanker and Rank1 cannot be attributed to potential differences in training conditions.

On MS MARCO, we find that StandardRanker outperforms Rank1 by an average of 3.7, 5, and 5.7 points for the 3B, 7B, and 14B model sizes, respectively. These gains are consistent, with StandardRanker being more effective on each individual dataset. On BRIGHT — a benchmark built on reasoning-intensive queries — we find a similar story where removing reasoning improves accuracy: StandardRanker outperforms Rank1 (our impl.), achieving 1, 3.2, and 5.2 points higher average nDCG@10 across the 3B, 7B, and 14B model sizes. Surprisingly, this effectiveness gap appears to *increase* as the backbone LLM gets larger, suggesting that scaling up benefits StandardRanker more than Rank1. At the 3B and 7B scale, Rank1 is consistently stronger on the AoPS and TheoQ. datasets,

---

[3]These LoRA parameters are consistent with those used in Weller et al. (2025).

| | StackExchange | | | | | | | Coding | | Theorem-based | | | Avg. |
|---|---|---|---|---|---|---|---|---|---|---|---|---|---|
| | Bio. | Earth. | Econ. | Psy. | Rob. | Stack. | Sus. | Leet. | Pony | AoPS | TheoQ. | TheoT. | |
| BM25 + GPT-4 CoT | 53.6 | 54.1 | 24.3 | 38.7 | 18.9 | 27.7 | 26.3 | 19.3 | 17.6 | 3.9 | 19.2 | 20.8 | 27.0 |
| **+ Qwen2.5-3B** | | | | | | | | | | | | | |
| StandardRanker | 41.6 | 27.1 | **20.9** | 31.9 | **22.2**$^\dagger$ | 16.9 | **30.3** | 13.2 | **42.0**$^\dagger$ | 2.7 | 16.2 | 30.6 | **24.6** |
| Rank1 (our impl.) | 37.3 | **27.8** | 20.7 | **33.1** | 18.3 | **24.3** | 25.2 | 11.3 | 26.2 | **4.7** | 20.7 | **34.0** | 23.6 |
| Rank1 | **41.8** | 25.6 | 18.4 | 29.3 | 15.5 | 18.4 | 25.8 | **16.1** | 24.9 | 4.7$^\ddagger$ | **21.7**$^\ddagger$ | 32.7 | 22.9 |
| **+ Qwen2.5-7B** | | | | | | | | | | | | | |
| StandardRanker | 47.1 | **38.0** | **28.1**$^\dagger$ | **44.1**$^\dagger$ | **26.1**$^\dagger$ | **29.5**$^\dagger$ | **36.5**$^\dagger$ | **19.3**$^\dagger$ | **37.5**$^\dagger$ | 4.6 | 22.4 | **39.4** | **31.0** |
| Rank1 (our impl.) | 47.0 | 35.4 | 24.0 | 35.2 | 20.0 | 25.2 | 31.0 | 15.1 | 36.0 | 5.9 | 22.2 | 36.6 | 27.8 |
| Rank1 | **48.8** | 36.7 | 20.8 | 35.0 | 22.0 | 18.7 | 36.2 | 12.7 | 31.2 | **6.3** | **23.7**$^\ddagger$ | 37.8 | 27.5 |
| **+ Qwen2.5-14B** | | | | | | | | | | | | | |
| StandardRanker | **52.9** | **45.5** | **30.6** | **46.1** | **28.5** | **32.3** | **38.1** | **24.1** | 33.1 | 8.0 | **26.8** | 40.7 | **33.9** |
| Rank1 | 49.3 | 37.7 | 22.6 | 35.2 | 22.5 | 20.8 | 33.6 | 17.7 | **33.2** | **8.4** | 22.5 | 41.4 | 28.7 |

Table 2: Out-of-domain performance of StandardRanker versus Rank1. Each Qwen2.5 model reranks the top-100 passages from BM25 with GPT-4 CoT query expansions (BM25 + GPT-4 CoT). **Bold** denotes best reranker under each Qwen2.5 model. GPT-4 CoT expansions are *not* used when reranking. $^\dagger$ denotes a statistically significant improvement over Rank1 based on a paired students t-test ($p < 0.05$). Similarly, $^\ddagger$ denotes a statistically significant improvement over StandardRanker.

| Qwen2.5 | Method | MS MARCO | BRIGHT |
|---|---|---|---|
| 3B | Rank1 (our impl.) | 57.8 | 23.6 |
| | Rank1-NoReason | **58.3** | 23.4 |
| | Rank1 | 57.2 | 22.9 |
| | Rank1-NoReason | 55.0 | **24.2** |
| 7B | Rank1 (our impl.) | 57.4 | 27.8 |
| | Rank1-NoReason | 58.3 | 30.8 |
| | Rank1 | 57.1 | 27.5 |
| | Rank1-NoReason | **58.6** | **31.5** |
| 14B | Rank1 | 56.2 | 28.7 |
| | Rank1-NoReason | **57.3** | **34.7** |

Table 3: Studying the effect of removing the reasoning process from Rank1. Results on MS MARCO and BRIGHT represent an average across the corresponding datasets. **Bold** results denote best between Rank1 and Rank1-NoReason. See Appendix F for results on individual datasets.

which are math-based tasks to retrieve relevant passages based theorem-based questions. This result is consistent with Sprague et al. (2025), who found that reasoning is most helpful on problems which require mathematical reasoning. However, this gap appears to close at the 14B reranker scale.

All in all, these results suggest that training a Rank1-style pointwise reranker does not provide any general advantage versus StandardRanker.

## 3.2 RQ2: How Important is the Reasoning Process to Rank1?

Our results up to this point demonstrate that, under the exact same training regime, rerankers that are trained to simply output a relevance prediction (StandardRanker) outperform rerankers trained to reason prior to making the relevance prediction (Rank1), on average. But, *what if we disable the reasoning for Rank1?* We hypothesize that if the reasoning is crucial to Rank1's relevance prediction, its reranking accuracy should drop if it does not reason.

**Experiment Setup.** In order to disable the reasoning for Rank1 (Rank1-NoReason), we pre-fill the LLM's reasoning with a dummy reasoning process: `<think> Okay, I have finished thinking. </think>` following the setup from Ma et al. (2025). We then follow the same evaluation setup as in Section 3.1. Note that this, in essence, turns Rank1 into a standard pointwise reranker as it only needs to output the relevance label.

**Results.** The results of this experiment can be found in Table 3. Across the 7B and 14B Rank1 models, Rank1-NoReason is consistently more effective than Rank1 on both MS MARCO and BRIGHT, with up to 1.5 and 3-point gains for the 7B model, respectively, and 1.1 and 6-point improvement for the 14B model, respectively. For the 3B model, the gains from Rank1-NoReason are less consistent, but we do find that Rank1-NoReason is generally around the same average nDCG@10 as Rank1. Similar to the result shown in Section 3.1, as Rank1's size increases, the benefits of removing the reasoning process *increase*. In fact, on the BRIGHT dataset, Rank1-NoReason at the 7B and 14B model size is on par with StandardRanker in terms of effectiveness, on average.

How does the text used for the reasoning process impact Rank1-NoReason's accuracy? In Table 4, we study different dummy reasoning variations for Rank1-NoReason (14B) on MS MARCO. The results show that completely leaving the reasoning blank (i.e., `<think></think>`) results in the largest drop in effectiveness compared to Rank1 (56.2 vs. 55.7). On the other hand, simply repeating the query-passage pair (`<think>{query}\n{passage}</think>`) results in a 1.9-point gain in nDCG@10, on average. In other words, repeating the input in the reasoning chain is more helpful than the reasoning chain generated by Rank1.

| Reasoning Process | MS MARCO |
|---|---|
| Rank1-14B | 56.2 |
| *Okay, I have finished thinking* | 57.3 |
| *Blank (empty string)* | 55.7 |
| *Passage* | 56.6 |
| *Query & Passage* | **58.1**[†] |

Table 4: Variations of the dummy reasoning process for Rank1-NoReason (14B). [†] denotes a statistically significant improvement over Rank1 based on a paired students t-test ($p < 0.05$).

The results in this section suggest that it is in fact the generated reasoning process that is limiting Rank1's effectiveness: If Rank1-NoReason at larger model sizes is competitive with StandardRanker, this means Rank1 is in fact learning how to predict relevance, but the reasoning it generates is not guiding it towards *better reranking ability*. We explore potential explanations in the next section.

## 4 Why Does Reasoning Hurt LLM Pointwise Rerankers?

One reason why Rank1 may perform worse than StandardRanker is that Rank1 has poorly calibrated and polarized probabilities for ranking due to the conclusions made by its reasoning process. For example, Rank1 will almost always assign very high probabilities when its reasoning concludes that a passage is relevant, and thus may not be able to reflect that one passage may be *more* relevant than another passage. On the other hand, as StandardRanker is trained to only output "true" or "false", it may implicitly learn to output scores that account for one passage being more relevant than another passage. Due to this, we hypothesize that StandardRanker can better model the *partial* relevance of query-passage pairs, making the outputs less polarized and preserving the uncertainty of scores which can be essential for the effectiveness of pointwise rerankers.

In this section, we dive deeper into this hypothesis. First, we investigate how Rank1 compares to StandardRanker and Rank1-NoReason as a simple binary relevance classifier. Then, we compare the relevance score distributions for StandardRanker, Rank1, and Rank1-NoReason. Lastly, we examine a qualitative example of Rank1's reasoning process.

### 4.1 Relevance Classification Comparison

We first study how different reranking methods compare as simple relevance classifiers, ignoring their reranking accuracy measured by metrics like nDCG@10, which, ultimately, is what we *really* care about. Doing so will allow us to better understand how much we can attribute differences in effectiveness to simply being worse relevance classifiers. In most cases, better relevance classification *should* result in better reranking accuracy.

For this experiment, we set $y_{pred} = 1$ if $R > 0.5$, and $y_{pred} = 0$ otherwise. For the ground truth relevance judgements, we set judgements > 2 (corresponding to highly relevant and perfectly relevant) as positive labels, and the rest as negative labels, following standard practice used for binary measures in IR (MacAvaney et al., 2022). The results for the Qwen2.5-7B models are in Table 5.

Comparing StandardRanker to Rank1, we find that in terms of F1-score and precision, StandardRanker is consistently stronger than Rank1. However, Rank1 generally has higher recall than StandardRanker,

| | MS MARCO v1 | | | | | | MS MARCO v2 | | | | | | | | |
| | DL19 | | | DL20 | | | DL21 | | | DL22 | | | DL23 | | |
| | P | R | F1 | P | R | F1 | P | R | F1 | P | R | F1 | P | R | F1 |
|---|---|---|---|---|---|---|---|---|---|---|---|---|---|---|---|
| StandardRanker | **71.4** | 80.3 | **75.6** | **54.5** | 79.3 | **64.6** | **56.4** | 87.1 | **68.5** | **49.1** | 70.0 | **57.6** | **39.7** | 66.9 | **49.8** |
| Rank1 | 65.9 | 82.4 | 73.2 | 49.2 | 82.1 | 61.5 | 54.0 | 89.2 | 67.3 | 41.6 | 73.2 | 53.1 | 35.6 | 61.9 | 45.2 |
| + Self-Consistency | 65.7 | **85.5** | 74.3 | 49.1 | 84.0 | 62.0 | 53.5 | 90.1 | 67.2 | 43.0 | 76.0 | 54.9 | 36.0 | 66.5 | 46.8 |
| Rank1-NoReason | 60.2 | 84.2 | 70.2 | 44.7 | **84.8** | 58.6 | 52.0 | **92.7** | 66.6 | 36.6 | **79.2** | 50.1 | 34.1 | **73.9** | 46.7 |

Table 5: Comparison of relevance classification performance (Precision, Recall, and F1-score) across StandardRanker and Rank1 (our impl.) variants. Self-Consistency will be discussed in Section 5.

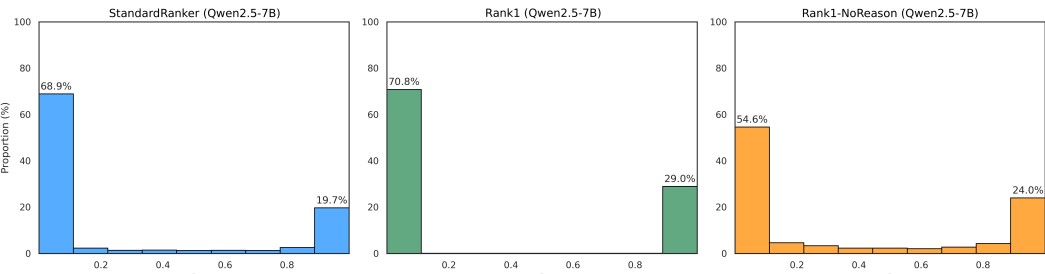

Figure 2: Relevance Scores Distribution across StandardRanker and Rank1 (our impl.) variants on DL19 with Qwen2.5-7B.

indicating that Rank1 is classifying passages as relevant more frequently. We note that this is further confirmed in Figure 2, which we discuss in the next subsection. Surprisingly, Rank1-NoReason is generally worse at relevance classification than Rank1 (in terms of F1 and precision), yet outperforms it in terms of retrieval metrics, as discussed in Section 3.2. Over the next two subsections, we provide potential explanations for this observation.

## 4.2 RELEVANCE SCORES DISTRIBUTION

Our observations in Section 4.1 revealed a mismatch between relevance classification precision and reranking accuracy metrics (i.e., nDCG@10) for Rank1 versus Rank1-NoReason. To better understand why this may be the case, we plot the distribution of the relevance scores across the Qwen2.5-7B rerankers, shown in Figure 2.

We find that StandardRanker and Rank1 place a similar proportion of their predictions in the low-relevance bin (0–0.1) for around 70% of its scores. However, while StandardRanker spreads its remaining scores across both partial-relevance (0.1–0.9) regions (11.4%) and high-relevance (0.9–1.0) regions (19.7%), Rank1 exhibits a more extreme distribution. It places almost no scores in intermediate regions, assigning essentially any passage with partial relevance exclusively into the very high relevance bins (29.0%). This demonstrates that the reasoning process leads the model to make polarized decisions, either relevant or not relevant, and rarely assigning partial relevance, matching our hypothesis.

Interestingly, if we take Rank1 and forcefully remove its reasoning process as done in Section 3.2 (i.e., Rank1-NoReason), this no longer becomes the case. While the majority of the relevance scores are still at the tails of the distribution, Rank1 begins to place more of its predictions across the partial-relevance bins (21.4%), while placing less predictions in the low-relevance and high-relevance bins.

## 4.3 QUALITATIVE STUDY

Finally, to better understand how Rank1 handles partially relevant passages, we examine the reasoning process of Rank1 for a random example in which StandardRanker produces a partially relevant prediction (i.e., in the 0.1-0.9 bin). The qualitative example can be found in Table 6. Looking at the reasoning process, we find that Rank1 explicitly mentions that the passage is "somewhat relevant", but because it has to select between two binary options (relevant or not relevant), it reasonably selects the passage as relevant. This subsequently makes $R$ high (0.999) as Rank1 finishes its reasoning

| Query | how to help a jammed finger |
|---|---|
| Passage | Jammed finger vs. broken finger. Share on Pinterest. A broken finger is often more painful than a jammed finger, although both may be treated using a splint. A broken finger will cause severe pain and swelling that lasts for hours or even days. While a jammed finger is painful, it is not usually severe. |
| Reasoning | Okay, let's see. The user is asking how to help a jammed finger. The passage provided talks about the difference between a jammed finger and a broken finger. It mentions that both might be treated with a splint, but a broken finger is more painful and has more swelling. First, the query is about treatment for a jammed finger. The passage does mention that a jammed finger can be treated with a splint, which is a direct answer to the query. However, the main focus of the passage is comparing jammed and broken fingers, not giving detailed treatment steps. It does say that both might use a splint, which is relevant. But the passage doesn't go into other treatment methods like icing, elevation, or when to see a doctor. So, the passage is somewhat relevant because it mentions a treatment (splint) for a jammed finger. But it's not comprehensive. The user might need more information, but the passage does provide a basic answer. However, the question is whether the passage is relevant, not if it's complete. Since it does address treatment (splint) for a jammed finger, even briefly, it's relevant. The comparison to a broken finger might also help in understanding when a jammed finger isn't severe. Therefore, the answer is true. |
| $R$ | **StandardRanker**: 0.374 
 **Rank1**: 0.999 
 **Rank1-NoReason**: 0.810 |

Table 6: An example of Rank1's (Qwen2.5-7B) reasoning process where it scores a partially relevant passage as highly relevant. As reference, we additionally provide the probability of relevance, $R$ for the provided query-passage pair. Appendix L provides additional examples.

stating "Therefore, the answer is true". This example shows how the reasoning process can bottleneck the ability for Rank1 to measure any concept of partial relevance, even when the model states that the passage is not fully addressing the query.

## 4.4 TAKEAWAYS

The findings from Sections 4.1 and 4.2 suggest that the ability to assign partial relevance is critical to the effectiveness of pointwise rerankers, not just the ability to predict if a passage is relevant or not. While Rank1 achieves higher effectiveness on binary relevance classification metrics compared to Rank1-NoReason, it still falls behind Rank1-NoReason in terms of nDCG@10. To maximize nDCG@10, it requires that rankers can properly capture the different degrees of relevance (placing the most relevant documents ahead of documents that are somewhat relevant or irrelevant). To quantify this observation and further confirm that partial relevance modeling is important to nDCG, we fit the following regression on the per-query nDCG@10 results for StandardRanker (7B), Rank1 (7B) and Rank1-NoReason (7B) across all MS MARCO datasets (DL19, DL20, DL21, DL22, DL23):

$$ndcg@10 = \underset{(p<0.001)}{0.5641} + \underset{(p<0.001)}{0.0449} \cdot \text{Score Gap} + \underset{(p<0.001)}{0.1845} \cdot P@10$$

Here, Score Gap denotes the average difference of $R$ assigned to true positives versus false positives. A higher Score Gap indicates that the reranker does a better job at capturing *relative* relevance (i.e., distinguishing labeled relevant documents from irrelevant ones that were incorrectly classified as relevant; see Appendix D for why this is important). P@10 measures the precision, i.e., the number of correct predictions in the top-10. For this regression, Score Gap and P@10 are standardized. The R-Squared of the regression is 0.684, indicating that Score Gap and P@10 account for $\approx 68\%$ of the variability of nDCG@10.

As expected, we find that having higher precision contributes the most to a higher nDCG@10. However, we also find that when controlling for the P@10, the ability to better separate the scores of true positives from the false positives has a positive and significant effect. In other words, the ability to capture relative relevance is positively and significantly correlated with nDCG. This quantitatively

|  | MS MARCO | BRIGHT |
|---|---|---|
| StandardRanker (7B) | 62.4 | 31.0 |
| Rank1-NoReason (7B) | 58.8 | 30.8 |
| Rank1 (7B) | 57.4 | 27.8 |
| + Self-Consistency | 59.2 | **30.7** |
| w/ Inverse Training | **61.6** | 30.5 |

Table 7: Influence of Self-Consistency and Inverse Training on Rank1.

|  | BRIGHT |
|---|---|
| ReasonIR | 30.8 |
| + StandardRanker (14B) | 34.8 |
| + Rank-R1 (14B)‡ | 29.7 |
| + Rank-K (32B)† | 32.6 |
| + ReasonRank (32B)† | **38.0** |

Table 8: Comparison of StandardRanker to listwise† and setwise‡ reasoning rerankers.

confirms our hypothesis: As Rank1 is unable to differentiate the nuances in relevance (always assigning the same $R$ score to all positive predictions), it results in lower nDCG@10. We provide further explanation of this result in Appendix D.

## 5 DISCUSSION AND RELATED WORK

**Is improving Rank1 as simple as incorporating partial relevance?** A common approach to improve reasoning models and incorporate diversity into its outputs is via multiple inference calls to the reasoning model. In Table 7, we explore this via self-consistency (Wang et al., 2023), which we denote as Rank1 + Self-Consistency. Unlike the majority vote approach used by Wang et al. (2023), we average the predicted $R$ values across eight sampled outputs from Rank1 to produce a continuous score suitable for reranking. As shown in Figure 3, Rank1 begins to distribute its relevance scores away from the low-relevance (0–0.1) and high-relevance (0.9–1.0) bins. In turn, nDCG@10 improves by 1.8 points on MS MARCO and 2.9 points on BRIGHT, even though the relevance classification metrics presented in Table 5, particularly precision, is generally on par with Rank1. While this brings an improvement to Rank1, it is still equally as effective as simply injecting a dummy reasoning process into the same model (i.e., Rank1-NoReason), while being more compute-intensive. These results suggest naive inference-time solutions are not enough to improve Rank1.

**Can reasoning be helpful for pointwise rerankers?** We believe our study supports the following conclusion: At least in Rank1's current state, reasoning is not necessary, or useful, for pointwise rerankers at inference time. *But is there potential for it to help?* This is unclear. Past results have shown that the benefits of reasoning come primarily on tasks which involve math or logic and that CoT reasoning is in fact unnecessary for most tasks its applied to (Sprague et al., 2025). Other work in the IR community (Ferraretto et al., 2023) has demonstrated that training LLM-based pointwise rerankers with synthetic GPT CoT explanations *can* be helpful, but with a critical caveat: the reasoning needs to be produced *after* making the relevance prediction. They found that producing the

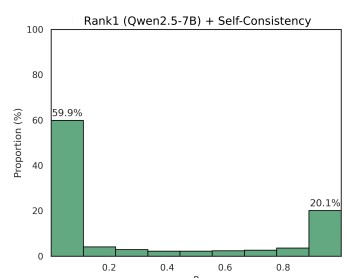

Figure 3: Relevance Scores Distribution for Rank1 + Self-Consistency on DL19.

reasoning first dropped nDCG@10 by 10 points. On the other hand, Samarinas & Zamani (2025) demonstrated encouraging results when using *discrete*, non-binary, labels: Could redesigning Rank1 to generate non-binary relevance labels be sufficient? We investigate a simple strategy to train Rank1 to produce non-binary labels in Appendix I. We find that while this underperforms Rank1 if the non-binary relevance labels are used directly as the relevance score (e.g., $R$), when the relevance labels are combined with first-stage retrieval scores (e.g., from BM25 + GPT-4 CoT), similar to what is done in Shao et al. (2025), using non-binary relevance labels can make a 3.3-point improvement in nDCG@10 to Rank1 on BRIGHT. However, it is *still* less effective than StandardRanker in the same hybrid setup. While promising, more work is needed to train a Rank1-style reranker that can produce non-binary relevance labels and is capable of outperforming StandardRanker. We leave this to future work.

**Can reasoning be useful as a data augmentation technique?** Rank1-NoReason's strong effectiveness raises an interesting question: Can reasoning be useful as an additional training signal,

where, for example, the reranker is trained to generate a reasoning process, but at inference time, the reasoning process gets "turned off"? Our results suggest this is a promising direction for future work, particularly for larger LLMs, as we found Rank1-NoReason to perform on par with StandardRanker at the 7B and 14B scales. Could larger models (e.g., 32B+) see even more improvement? Alternatively, what if Rank1 is trained to produce the reasoning chain *after* generating a relevance label (Ferraretto et al., 2023)? In Table 7, we test this "inverse training" approach: Rank1 is trained to first produce a relevance label followed by its reasoning process, but at inference time we only allow Rank1 to produce the relevance label. With inverse training, Rank1 can achieve similar effectiveness to StandardRanker on MS MARCO (61.6 vs. 62.0) and BRIGHT (30.5 vs. 31.0). One advantage of reasoning rerankers is their potential to improve explainability of ranking results (Weller et al., 2025; Zhuang et al., 2025). Rank1 with inverse training can serve as a middle ground between Rank1 and StandardRanker, where one gets the explainability benefits of a reasoning model (if, say, requested by the user), while preserving the stronger ranking capabilities of StandardRanker.

**What about reasoning for non-pointwise rerankers?**   Since the introduction of Rank1 (Weller et al., 2025), various follow-up methods have been developed, focusing on injecting a reasoning process into listwise (Liu et al., 2025; Yang et al., 2025b) and setwise (Zhuang et al., 2025) LLM rerankers. In Table 8, we compare StandardRanker to these rerankers. For a fair comparison to the results reported in Liu et al. (2025), we rerank the top-100 passages from ReasonIR (Shao et al., 2025), a dense retriever trained for reasoning tasks. How does StandardRanker compare? *Surprisingly well.* StandardRanker is only outperformed by ReasonRank, which leverages a sophisticated automated training data curation strategy to mine data from domains similar to those in BRIGHT. Compared to Rank-R1 and Rank-K, rerankers which were both trained on the MS MARCO dataset, StandardRanker performs 5.1 and 2.2 points better, on average. While not fully conclusive if reasoning is more helpful for non-pointwise rerankers, these results suggest that non-reasoning baselines remain strong and that it is imperative to properly evaluate them under equivalent settings. We leave this to future work.

## 6 CONCLUSION

In this work, we study whether the generation of a reasoning chain prior to making a relevance prediction actually improves the accuracy of pointwise rerankers. To do so, we compare Rank1 to two simple, but effective pointwise reranking baselines: StandardRanker and Rank1-NoReason. Through experiments across in-domain and out-of-domain datasets, we find that the reasoning process consistently *harms* the accuracy of pointwise rerankers, finding larger gaps in effectiveness as LLM size increases. Investigating the root cause of this result, we observe that the reasoning process restricts Rank1's ability to capture partial relevance between query-passage pairs, which is an important factor for pointwise reranking accuracy.

## ETHICS STATEMENT

Our research solely uses publicly available datasets, and no personal information is collected. All datasets and models are used in accordance with its intended use and licenses. The goal of our study is to better understand the factors that influence the accuracy of LLM pointwise rerankers, which we hope can have a positive impact on building better search engines and other applications built on retrieval systems.

While our results showed that standard pointwise rerankers, which minimize the output tokens generated by an LLM, outperform more verbose reasoning pointwise rerankers, we do recognize that such systems still rely on LLMs, which means that there is a risk that the LLM can produce biased, harmful, or offensive output.

## REPRODUCIBILITY STATEMENT

For reproducing our StandardRanker and Rank1 (our impl.) results, we share training details in the experiment setup paragraph in Section 3.1 and also provide more in-depth training details in Appendix E. For training Rank1 with non-binary relevance labels, we provide training details in Appendix I. Furthermore, we release all prompts used for training and inference in Appendix H for

StandardRanker, Rank1, Rank1-NoReason. To ensure our results are reproducible, we provide our source code in the supplementary materials and will be made public upon acceptance. Please follow the instructions in the README.md file to run the experiments.

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

## A  THE USE OF LARGE LANGUAGE MODELS (LLMs)

LLMs were solely utilized to assist in making grammatical edits for sections of this work, including the text and tables.

## B  DATASET DETAILS

We show the number of test queries for each dataset used for evaluation in Table 9.

| Dataset | # Queries |
|---|---|
| TREC DL19 | 43 |
| TREC DL20 | 54 |
| TREC DL21 | 53 |
| TREC DL22 | 76 |
| TREC DL23 | 82 |
| Biology | 103 |
| Earth Science | 116 |
| Economics | 103 |
| Psychology | 101 |
| Robotics | 101 |
| Stackoverflow | 117 |
| Sustainable Living | 108 |
| Leetcode | 142 |
| Pony | 112 |
| AoPs | 111 |
| TheoremQA Questions | 194 |
| TheoremQA Theorems | 76 |

Table 9: Dataset Details

## C  MODEL DETAILS

- `Qwen2.5-3B`: A 3B base model. Huggingface ID: `Qwen/Qwen2.5-3B`
- `Qwen2.5-7B`: A 7B base model. Huggingface ID: `Qwen/Qwen2.5-7B`
- `Qwen2.5-14B`: A 14B base model. Huggingface ID: `Qwen/Qwen2.5-14B`
- `rank1-3b`: A 3B pointwise reasoning reranker. Huggingface ID: `jhu-clsp/rank1-3b`
- `rank1-7b`: A 7B pointwise reasoning reranker. Huggingface ID: `jhu-clsp/rank1-7b`
- `rank1-14b`: A 14B pointwise reasoning reranker. Huggingface ID: `jhu-clsp/rank1-14b`

## D  WHY IS MODELING PARTIAL OR RELATIVE RELEVANCE IMPORTANT FOR RERANKERS?

In Section 4, our results suggest that the inability to model partial relevance is a reason why Rank1 underperforms StandardRanker. To better explain this, we first define what we mean by partial relevance: In ranking, the ideal ranked list is one in which all the relevant documents are ranked ahead of the irrelevant ones. For example, imagine relevance is judged on a 4-point scale, like those of MS MARCO: Perfectly Relevant, Highly Relevant, Related, Irrelevant. To maximize nDCG, the ideal LLM reranker would return a list such that Perfectly Relevant > Highly Relevant > Related > Irrelevant. What our results hint is that when Rank1 makes a relevance prediction, regardless of if the passage is Perfectly Relevant or Related, it returns the same $R$, whereas StandardRanker does a better job implicitly *differentiating* between the scores.

To empirically demonstrate this, in Table 12, we show the average $R$ prediction for each relevance class when the reranker classifies a document as *relevant (i.e., true)*. As we see, when Rank1 predicts that a document is relevant, it always produces the same $R$ value (i.e., 0.999), regardless of whether

|  |  | DL19 | DL20 | DL21 | DL22 | DL23 | Avg. |
|---|---|---|---|---|---|---|---|
| 3B | StandardRanker | 72.5 | 68.9 | 69.4 | 51.4 | 45.5 | 61.5 |
|  | Rank1 (our impl.) | 70.4 | 66.4 | 65.9 | 45.2 | 41.3 | 57.8 |
|  | NoReason | 71.8 | 63.7 | 66.8 | 47.1 | 41.9 | 58.3 |
|  | Rank1 | 70.1 | 64.5 | 65.3 | 45.2 | 40.7 | 57.2 |
|  | NoReason | 69.4 | 62.3 | 63.3 | 43.1 | 37.1 | 55.0 |
| 7B | StandardRanker | 74.6 | 70.0 | 70.9 | 50.3 | 46.3 | 62.4 |
|  | Rank1 (our impl.) | 70.3 | 64.3 | 65.9 | 45.6 | 41.1 | 57.4 |
|  | NoReason | 73.3 | 65.0 | 69.1 | 46.1 | 40.5 | 58.8 |
|  | + Self-Consistency | 71.5 | 66.7 | 68.8 | 46.0 | 42.9 | 59.2 |
|  | w/ Inverse Training | 74.3 | 69.3 | 70.5 | 50.4 | 43.4 | 61.6 |
|  | Rank1 | 68.4 | 65.1 | 65.8 | 44.7 | 41.6 | 57.1 |
|  | NoReason | 74.2 | 64.0 | 67.8 | 46.9 | 40.0 | 58.6 |
| 14B | StandardRanker | 73.3 | 68.7 | 70.7 | 49.6 | 47.3 | 61.9 |
|  | Rank1 | 66.3 | 65.0 | 63.1 | 44.8 | 41.7 | 56.2 |
|  | NoReason | 70.2 | 62.3 | 66.2 | 45.5 | 42.4 | 57.3 |
|  | NoReason: *Blank (empty string)* | 69.4 | 61.1 | 64.8 | 42.9 | 40.2 | 55.7 |
|  | NoReason: *Passage* | 68.2 | 64.2 | 64.9 | 44.8 | 40.8 | 56.6 |
|  | NoReason: *Query & Passage* | 68.4 | 65.4 | 66.7 | 46.6 | 43.2 | 58.1 |

Table 10: Full results for StandardRanker, Rank1, Rank1-NoReason, Rank1 + Self-Consistency, and Rank1 w/ Inverse Training. All models rerank the top-100 passages from BM25.

|  |  | Bio. | Earth. | Econ. | Psy. | Rob. | Stack. | Sus. | Leet. | Pony | AoPS | TheoQ. | TheoT. | Avg. |
|---|---|---|---|---|---|---|---|---|---|---|---|---|---|---|
| 3B | StandardRanker | 41.6 | 27.1 | 20.9 | 31.9 | 22.2 | 16.9 | 30.3 | 13.2 | 42.0 | 2.7 | 16.2 | 30.6 | 24.6 |
|  | Rank1 (our impl.) | 37.3 | 27.8 | 20.7 | 33.1 | 18.3 | 24.3 | 25.2 | 11.3 | 26.2 | 4.7 | 20.7 | 34.0 | 23.6 |
|  | NoReason | 40.8 | 20.5 | 20.3 | 31.9 | 14.0 | 15.3 | 23.3 | 18.7 | 37.3 | 3.7 | 24.6 | 31.1 | 23.4 |
|  | Rank1 | 41.8 | 25.6 | 18.4 | 29.3 | 15.5 | 18.4 | 25.8 | 16.1 | 24.9 | 4.7 | 21.7 | 32.7 | 22.9 |
|  | NoReason | 40.6 | 20.9 | 19.9 | 32.8 | 17.8 | 16.3 | 25.6 | 19.0 | 38.0 | 2.4 | 25.2 | 31.8 | 24.2 |
| 7B | StandardRanker | 47.1 | 38.0 | 28.1 | 44.1 | 26.1 | 29.5 | 36.5 | 19.3 | 37.5 | 4.6 | 22.4 | 39.4 | 31.0 |
|  | Rank1 (our impl.) | 47.0 | 35.4 | 24.0 | 35.2 | 20.0 | 25.2 | 31.0 | 15.1 | 36.0 | 5.9 | 22.2 | 36.6 | 27.8 |
|  | NoReason | 56.0 | 41.9 | 27.5 | 38.5 | 23.2 | 21.6 | 32.7 | 16.3 | 39.4 | 7.2 | 27.2 | 38.0 | 30.8 |
|  | + Self-Consistency | 49.6 | 38.2 | 27.4 | 40.9 | 23.7 | 29.3 | 33.2 | 14.9 | 38.4 | 8.1 | 25.4 | 39.1 | 30.7 |
|  | w/ Inverse Training | 48.3 | 43.5 | 27.7 | 39.3 | 23.9 | 26.4 | 34.6 | 21.7 | 28.8 | 5.1 | 28.8 | 37.5 | 30.5 |
|  | Rank1 | 48.8 | 36.7 | 20.8 | 35.0 | 22.0 | 18.7 | 36.2 | 12.7 | 31.2 | 6.3 | 23.7 | 37.8 | 27.5 |
|  | NoReason | 52.1 | 40.4 | 29.4 | 43.1 | 26.1 | 27.3 | 32.5 | 18.3 | 35.1 | 6.8 | 28.8 | 38.7 | 31.5 |
| 14B | StandardRanker | 52.9 | 45.5 | 30.6 | 46.1 | 28.5 | 32.3 | 38.1 | 24.1 | 33.1 | 8.0 | 26.8 | 40.7 | 33.9 |
|  | Rank1 | 49.3 | 37.7 | 22.6 | 35.2 | 22.5 | 20.8 | 33.6 | 17.7 | 33.2 | 8.4 | 22.5 | 41.4 | 28.7 |
|  | NoReason | 59.8 | 44.9 | 31.2 | 46.4 | 26.7 | 29.6 | 36.3 | 24.2 | 46.3 | 8.1 | 23.6 | 40.9 | 34.7 |
|  | NoReason: *Blank (empty string)* | 54.8 | 42.3 | 27.1 | 45.2 | 25.1 | 17.9 | 34.1 | 21.5 | 26.4 | 8.5 | 24.6 | 40.2 | 30.6 |

Table 11: Full results for StandardRanker, Rank1, Rank1-NoReason, Rank1 + Self-Consistency, and Rank1 w/ Inverse Training. All models rerank the top-100 passages from BM25 + GPT-4 CoT.

the document is relevant or not relevant. On the other hand, StandardRanker and Rank1-NoReason show a linear decrease in average $R$ when the document is Perfectly Relevant versus Highly Relevant, Related, and Irrelevant. Meaning when they score a document as "true", the order naturally places Perfectly Relevant ahead of Highly Relevant, and so on. On the other hand, Rank1 does not do so, meaning it orders documents randomly, not guaranteeing that Perfectly Relevant is ahead of Irrelevant, etc. Another way to look at this is as follows: When StandardRanker makes a misclassification (i.e., Irrelevant gets 0.953 in Table 12), on average, it will appear behind actual relevant documents (e.g., Related gets 0.974 in Table 12). Rank1 does not have this guarantee (i.e., both Irrelevant and Related get 0.999).

Now, one might ask: *But why does Rank1-NoReason perform worse than StandardRanker*? As we show in Section 4, this can be explained by its worse classification accuracy, which can be due to the train-test mismatch. Rank1-NoReason is trained with reasoning but tested without reasoning, while StandardRanker is consistently trained and tested without reasoning.

| | Relevance Label | | | |
|---|---|---|---|---|
| | **Irrelevant** | **Related** | **Highly Relevant** | **Perfectly Relevant** |
| **Rank1 (14B)** | 0.999 | 0.999 | 0.999 | 0.999 |
| **StandardRanker (14B)** | 0.953 | 0.974 | 0.982 | 0.993 |
| **Rank1-NoReason (14B)** | 0.930 | 0.963 | 0.987 | 0.992 |

Table 12: Average $R$ score when the LLM reranker predicts a given passage is relevant conditioned on its *true* label on DL19.

| | MS MARCO | BRIGHT |
|---|---|---|
| StandardRanker | 62.4 | 31.0 |
| Rank1-NoReason | 58.8 | 30.8 |
| Rank1 (our impl.) | 57.4 | 27.8 |
| + Self-Consistency (3 samples) | 59.1 | 30.6 |
| + Self-Consistency (8 samples) | 59.2 | 30.7 |

Table 13: Influence of the number of sampled chains for Rank1 + Self-Consistency.

# E   TRAINING AND INFERENCE DETAILS FOR STANDARDRANKER, RANK1, RANK1 + SELF CONSISTENCY, AND RANK1 W/ INVERSE TRAINING

To train StandardRanker and Rank1 (our impl.) we fine-tune Qwen2.5 using LoRA (Hu et al., 2022) for one epoch with rank 32 and alpha 64, using a batch size of 128 and a learning rate of 2e-4. We apply LoRA to all the linear layers of the transformer model. Note, to train the StandardRanker we leverage the same dataset as Rank1, but only use the (query, passage, relevance label) triples, ignoring the R1 reasoning process. To train Rank1 w/ Inverse Training, we follow the same exact setup as Rank1 (our impl.), but switch around the order of relevance label and R1's reasoning chain. In other words, we train with the following quadruples: (query, passage, relevance label, R1's reasoning chain). Training for each reranker took less than a day and was done on an A100 GPU. Due to limited computational resources, each model is only trained once.

For inference, we run all models on NVIDIA A6000 (48GB) and A100 (80GB) GPUs. As the StandardRanker and Rank1 outputs are run with greedy decoding, all the scores in the paper are from a single run. To run inference for Rank1 + Self Consistency, we sample eight outputs from Rank1 using a temperature of 0.7. Due to limited computational resources, we only run inference once per dataset. However, as we show in Section G, we found similar results when running Rank1 + Self Consistency when sampling three outputs versus eight outputs, suggesting results are generally consistent across runs.

# F   FULL RESULTS FOR RANK1-NOREASON, RANK1 + SELF-CONSISTENCY, AND RANK1 W/ INVERSE TRAINING

In this section, we provide the full results for Rank1-NoReason, Rank1 + Self-Consistency, and Rank1 w/ Inverse Training across MS MARCO and BRIGHT datasets. These results can be found Table 10 and Table 11.

# G   NUMBER OF SAMPLED OUTPUTS FOR RANK1 + SELF-CONSISTENCY

In this section, we present the results of Rank1 + Self-Consistency when we only sample 3 reasoning chains from Rank1. The results are shown in Table 13. We find that Rank1 + Self-Consistency (n=3) is as effective as Rank1 + Self-Consistency (n=8), suggesting that sampling more reasoning chains is not more effective for Rank1.

## H   PROMPTS

For training and evaluation of StandardRanker, and for our implementation of Rank1, and Rank1-NoReason, we leverage the exact same prompts used in the Rank1 (Weller et al., 2025) paper, but apply the Qwen2.5 (Yang et al., 2024) chat template. Below we repeat the baseline prompt. For the dataset specific prompts we used, please refer to Weller et al. (2025). Note for Rank1 with inverse training, we leverage the same prompt as StandardRanker.

**StandardRanker Prompt:**

```
<|im_start|>system
Determine if the following passage is relevant to the query. Answer only with 'true' or 'false'.
<|im_end|>
<|im_start|>user
Query: {}
Passage: {}
<|im_end|>
<|im_start|>assistant
```

**Rank1 (our impl.) Prompt:**

```
<|im_start|>system
Determine if the following passage is relevant to the query. Answer only with 'true' or 'false'.
<|im_end|>
<|im_start|>user
Query: {}
Passage: {}
<|im_end|>
<|im_start|>assistant
<think>
```

**Rank1-NoReason Prompt:**

```
<|im_start|>system
Determine if the following passage is relevant to the query. Answer only with 'true' or 'false'.
<|im_end|>
<|im_start|>user
Query: {}
Passage: {}
<|im_end|>
<|im_start|>assistant
<think>
Okay, I have finished thinking.
</think>
```

## I   TRAINING RANK1 TO PRODUCE NON-BINARY RELEVANCE LABELS

In this section, we explore a simple strategy to train Rank1 to produce graded, non-binary, relevance labels, rather than binary labels. We refer to this Rank1 variation as Rank1 (non-binary labels). To generate training data for Rank1 (non-binary labels), for each example, we first prompt Qwen2.5-7B-Instruct to generate a score from 0 to 4 given the query, passage and reasoning, using the following prompt:

```
<|im_start|>user
Is the document below relevant to answering the query below? The answer should be
'Relevance score: X.' where X is a number from 0-4, with each score defined below:
0 — Not Relevant: The document does not address the query at all.
1 — Marginally Relevant: The document touches on the topic very lightly but provides
almost no useful information.
2 — Partially Relevant: The document addresses some aspects of the query but is incomplete
or superficial.
3 — Mostly Relevant: The document addresses the query well but may miss minor points or
lack full detail.
4 — Highly Relevant: The document thoroughly addresses the query, providing detailed and
accurate information.
Here is the query:
<start_query>
{}
<end_query>

Here is the document:
<start_document>
{}
<end_document>

Here is your reasoning about the relevance of the document to the query:
<start_reasoning>
{}
<end_reasoning>
<|im_end|>
<|im_start|>assistant
Relevance score:
```

This in turn generates graded relevance labels for each of Rank1's training examples. We then train Rank1 as exactly as described in Appendix E, using the Qwen2.5-7B-Instruct graded labels rather than the original binary labels. Note that we *do not* adapt the reasoning chains provided in Rank1's training data and instead leave them fixed. We, however, think this could be a good direction for future work.

The results for this experiment can be found in Table 14. The results show that Rank1 (non-binary labels) is less effective than Rank1 when directly ranking with the generated non-binary relevance labels, likely due to there being many ties with passages being assigned the exact same label. To break ties, we investigate the effectiveness of the hybrid scoring approach utilized in Shao et al. (2025), which combines scores from BM25 + GPT-4 CoT with the generated relevance labels from Rank1 (non-binary labels) by interpolating them with a ratio of 0.5 (please refer to Shao et al. (2025) for more details). Using the hybrid scoring strategy, Rank1 (non-binary labels) improves in nDCG@10 by 8.6 points, on average and is even more effective that Rank1 under the same hybrid scoring strategy.

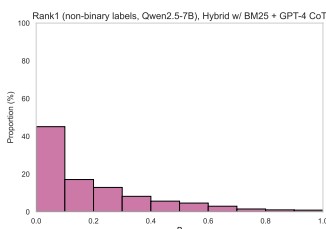

Figure 4: Relevance Scores Distribution for Rank1 (non-binary labels, Hybrid) on DL19.

StandardRanker still remains more effective than Rank1 under equal conditions (34.2 vs. 33.4). We include a plot of the distribution of Rank1 (non-binary labels) with the hybrid scoring in Figure 4. As we see the scores a distributed in a much less bimodal way, likely due to the scores from BM25 + GPT-4 CoT helping break ties.

| | StackExchange | | | | | | | Coding | | Theorem-based | | | Avg. |
|---|---|---|---|---|---|---|---|---|---|---|---|---|---|
| | Bio. | Earth. | Econ. | Psy. | Rob. | Stack. | Sus. | Leet. | Pony | AoPS | TheoQ. | TheoT. | |
| BM25 + GPT-4 CoT | 53.6 | 54.1 | 24.3 | 38.7 | 18.9 | 27.7 | 26.3 | 19.3 | 17.6 | 3.9 | 19.2 | 20.8 | 27.0 |
| + Qwen2.5-7B | | | | | | | | | | | | | |
| StandardRanker | 47.1 | 38.0 | 28.1 | 44.1 | 26.1 | 29.5 | 36.5 | 19.3 | 37.5 | 4.6 | 22.4 | 39.4 | 31.0 |
| Hybrid (w/ BM25 + GPT-4 CoT) | 56.3 | 52.9 | 31.6 | 45.7 | 27.0 | 33.6 | 36.8 | 26.7 | 35.0 | 5.2 | 22.9 | 37.0 | 34.2 |
| Rank1 (our impl.) | 47.0 | 35.4 | 24.0 | 35.2 | 20.0 | 25.2 | 31.0 | 15.1 | 36.0 | 5.9 | 22.2 | 36.6 | 27.8 |
| Hybrid (w/ BM25 + GPT-4 CoT) | 51.2 | 47.4 | 23.4 | 41.6 | 23.4 | 28.7 | 31.4 | 20.2 | 27.4 | 5.5 | 24.2 | 35.5 | 30.0 |
| Rank1 (non-binary labels) | 34.8 | 36.1 | 20.6 | 30.8 | 20.9 | 24.4 | 30.1 | 15.9 | 23.1 | 7.0 | 18.6 | 36.0 | 24.8 |
| Hybrid (w/ BM25 + GPT-4 CoT) | 53.4 | 54.3 | 29.2 | 44.9 | 27.1 | 32.1 | 35.8 | 22.1 | 32.3 | 8.6 | 24.2 | 36.5 | 33.4 |
| + Qwen2.5-32B-Instruct | | | | | | | | | | | | | |
| QwenRerank (Shao et al., 2025) | 63.6 | 59.2 | 30.2 | 45.7 | 29.6 | 33.6 | 33.7 | 27.9 | 29.0 | 6.3 | 24.1 | 35.7 | 34.9 |

Table 14: Training Rank1 to produce non-binary relevance labels.

| | First-Stage Retriever | |
|---|---|---|
| | BM25 + GPT-4 CoT | ReasonIR + GPT-4 CoT |
| StandardRanker | 31.0 | 31.2 |
| Rank1 (Our Impl.) | 27.8 | 25.8 |

Table 15: Impact of first-stage retriever on effectiveness of StandardRanker versus Rank1 on BRIGHT.

| | MS MARCO v1 | | MS MARCO v2 | | | Avg. |
|---|---|---|---|---|---|---|
| | DL19 | DL20 | DL21 | DL22 | DL23 | |
| BM25 | 50.6 | 48.0 | 44.6 | 26.9 | 26.3 | 39.3 |
| + Qwen2.5-3B | | | | | | |
| StandardRanker | **72.5** | **68.9** | **69.4** | **51.4** | **45.5** | **61.5** |
| Rank1 (our impl.) | 70.4 | 66.4 | 65.9 | 45.2 | 41.3 | 57.8 |
| + Qwen3-4B | | | | | | |
| StandardRanker | **72.7** | **69.0** | **70.1** | **50.0** | **45.8** | **61.5** |
| Rank1 (our impl.) | 67.7 | 65.4 | 64.8 | 44.1 | 42.2 | 56.8 |

Table 16: Investigating the generalizability of StandardRanker and Rank1 with Qwen3-4B.

## J    STANDARDRANKER AND RANK1 WITH DIFFERENT FIRST-STAGE RETRIEVERS

We explore how our results may be impacted when candidate documents are retrieved by different first-stage retrievers. The results for this experiment can be found in Table 15. Interestingly, StandardRanker remains stable in terms of nDCG@10, whereas Rank1 does worse.

## K    STANDARDRANKER AND RANK1 WITH DIFFERENT LLMS

We investigate the generalizability of our results by further testing StandardRanker and Rank1 with the Qwen3-4B. These results are found in Table 16. We find a similar result regardless of the LLM, with StandardRanker outperforming Rank1.

## L    ADDITIONAL QUALITATIVE EXAMPLES

Below we provide additional examples of Rank1's reasoning chains for given query-passage pairs from MS MARCO. Similar to what we did in Section 4.3, we randomly select examples in which either StandardRanker or Rank1-NoReason produce a partially relevant prediction. Below we find a similar story to what we found in 4.3: Even when Rank1 is acknowledging that the passage is not fully answering the question, it still scores the passage as highly relevant (i.e., providing a relevance score of 0.999). This in turn makes it difficult for Rank1 to return a proper ranking of passages as the relevance score of a partially relevant passage would be *indistinguishable* to that of a highly relevant passage.

| Query | are naturalization records public information |
|---|---|
| **Passage** | Civil Records Definition. Civil records are a group of public records that pertain to civil registry records, civil family matters and non criminal civil offenses. These records vary a lot because of the nature of the information that is recorded. |
| **Reasoning** | Okay, let's tackle this query. The user is asking if naturalization records are public information. The passage provided talks about civil records being public, covering civil registry, family matters, and non-criminal offenses. First, I need to determine if naturalization records fall under the categories mentioned in the passage. Naturalization is the process of becoming a citizen, which is a civil matter. The passage mentions civil registry and family matters, which are part of civil records. However, the passage doesn't explicitly mention naturalization. Next, I should consider if naturalization is considered a non-criminal civil offense. Since it's a legal process, it's likely included in civil records. But the passage doesn't list specific examples, so there's some ambiguity. The key point is whether the passage's definition of civil records includes naturalization. Since the passage states that civil records are public and covers various non-criminal matters, and naturalization is a civil matter, it's reasonable to infer that naturalization records are public. Therefore, the passage is relevant because it provides the basis for determining that such records are public. |
| $R$ | **StandardRanker**: 0.002
**Rank1**: 0.999
**Rank1-NoReason**: 0.280 |

Table 17: An example of Rank1's (Qwen2.5-7B) reasoning process where it scores a partially relevant passage as highly relevant. As reference, we additionally provide the probability of relevance, $R$, for StandardRanker, Rank1, and Rank1-NoReason for the provided query-passage pair.

| Query | what can you do about discrimination in the workplace in oklahoma city |
|---|---|
| **Passage** | Share on Google Plus. People who are deaf or have a hearing impairment may be protected from workplace discrimination by the Americans with Disabilities Act (ADA). The ADA applies to private employers with at least 15 employees, as well as to state and local government employers.If you are covered by the ADA, an employer may not discriminate against you based on your hearing impairment.You are also entitled to a reasonable accommodation – changes to the job or workplace that will allow you to do your job – as long as it doesn't create an undue hardship for your employer.he ADA applies to private employers with at least 15 employees, as well as to state and local government employers. If you are covered by the ADA, an employer may not discriminate against you based on your hearing impairment. |
| **Reasoning** | Okay, let's tackle this query. The user is asking about what can be done about discrimination in the workplace in Oklahoma City. The passage provided talks about the Americans with Disabilities Act (ADA) and how it protects people with hearing impairments from discrimination, including the requirement for reasonable accommodations. First, I need to determine if the passage is relevant to the query. The query is about workplace discrimination in Oklahoma City. The passage specifically mentions the ADA, which applies to employers with 15 or more employees, including state and local government employers. Oklahoma City is a city within Oklahoma, which is a state. So, the ADA would apply here. The passage also discusses protections against discrimination based on hearing impairments and the right to reasonable accommodations. However, the query is general about discrimination, not specifically about hearing impairments. The passage only addresses one type of discrimination (hearing impairment) under the ADA. But since the ADA is a federal law that applies to employers in Oklahoma City, the information is relevant to the broader question of workplace discrimination. Even though the passage is specific to hearing impairments, it's part of the legal framework that addresses discrimination in the workplace, which the user is asking about. Therefore, the passage is relevant because it provides a legal recourse (ADA) for a form of workplace discrimination in Oklahoma City. |
| $R$ | **StandardRanker**: 0.43 
 **Rank1**: 0.995 
 **Rank1-NoReason**: 0.09 |

Table 18: An example of Rank1's (Qwen2.5-7B) reasoning process where it scores a partially relevant passage as highly relevant. As reference, we additionally provide the probability of relevance, $R$, for StandardRanker, Rank1, and Rank1-NoReason for the provided query-passage pair.

| Query | description of mesquite plant |
|---|---|
| **Passage** | Mesquite Tree Problems. Hunker may earn compensation through affiliate links in this story. Mesquite trees are one of the toughest trees that grow in the desert and are usually problem free. The root system of a mesquite has been known to extend 200 feet below ground and horizontally far beyond the plant canopy. |
| **Reasoning** | Okay, let's tackle this question. The user wants to know if the given passage is relevant to the query "description of mesquite plant." First, the query is asking for a description of the mesquite plant. That would typically include things like its physical characteristics, habitat, structure, maybe even its uses or problems it might have. Looking at the passage, it starts by mentioning "Mesquite Tree Problems" and talks about how mesquite trees are tough in the desert and usually problem-free. Then it goes into detail about the root system being extensive, both vertically and horizontally. So, the passage does provide some information about the mesquite tree, specifically its root system and its problem-free nature. However, the main focus here is on the problems (or lack thereof) and the root system. The query is a general request for a description, which might expect more about the plant's appearance, growth habits, maybe flowers, fruit, etc. The passage doesn't mention those aspects. It does mention the tree's toughness and problem-free status, which could be part of a description, but the majority of the content is about the root system. But wait, the user is asking if the passage is relevant. Even if it's not a complete description, does it still provide some relevant information? The root system is a part of the plant's structure, which is part of a description. However, the passage is more focused on problems and the root system rather than a comprehensive description. The initial part mentions they are problem-free, which is a characteristic. So maybe that's part of the description. But the key here is whether the passage addresses the query's need for a description. The answer might be that it's partially relevant because it does mention some aspects (root system, problem-free nature) but doesn't cover other key parts like physical structure, appearance, etc. However, the question is whether it's relevant, not if it's comprehensive. Since it does provide some descriptive elements, even if limited, it could be considered relevant. But the passage's main focus is on problems and the root system, which might be more about issues rather than a general description. The query is about a description, so maybe the problems part is less relevant. However, the root system is part of the plant's description. Hmm, I'm a bit torn. The passage does give some descriptive information (root system, problem-free), so maybe it's relevant. But if the query expects a more detailed description including other aspects, then maybe not. But the user is asking if the passage is relevant, not if it's a complete answer. Since it does provide some descriptive elements, even if limited, the answer would be true. But wait, the passage starts with "Mesquite Tree Problems," which might be more about issues rather than a general description. The first sentence says they are problem-free, which is a characteristic. The rest is about the root system. So, the passage does include some descriptive elements (root system, problem-free), so it's relevant. Therefore, the answer should be true. |
| $R$ | **StandardRanker**: 0.712
**Rank1**: 0.999
**Rank1-NoReason**: 0.179 |

Table 19: An example of Rank1's (Qwen2.5-7B) reasoning process where it scores a partially relevant passage as highly relevant. As reference, we additionally provide the probability of relevance, $R$, for StandardRanker, Rank1, and Rank1-NoReason for the provided query-passage pair.

| | | |
|---|---|---|
| **Query** | does light intensity or concentration of carbon dioxide have a higher rate of photosynthesis | |
| **Passage** | Carbon dioxide. As carbon dioxide concentrations increase, so too does the rate of photosynthesis until a certain point where the graph levels off. At lower carbon dioxide concentrations carbon dioxide is the limiting factor because an increase in carbon dioxide causes an increase in photosynthesis. At higher carbon dioxide concentrations (plateau of graph), further increasing the carbon dioxide concentration does not increase the rate of photosynthesis meaning another factor is limiting photosynthesis. | |
| **Reasoning** | Okay, let's tackle this question. The user is asking whether light intensity or the concentration of carbon dioxide has a higher rate of photosynthesis. The passage provided talks about carbon dioxide levels affecting photosynthesis. It mentions that as CO2 increases, the rate goes up until a point where it plateaus. Then, at higher CO2 levels, another factor becomes the limiting factor. So, the query is comparing two factors: light intensity and CO2 concentration. The passage only discusses CO2, explaining that up to a certain point, CO2 is the limiting factor, but after that, another factor (probably light) takes over. However, the passage doesn't mention light intensity at all. It only states that when CO2 levels are high, another factor (not named) limits photosynthesis. The user wants to know which factor has a higher rate. The passage confirms that CO2 is a limiting factor up to a point, implying that beyond that, light intensity becomes the limiting factor. But the passage doesn't directly compare the two factors or state which one is more influential overall. It just explains the role of CO2. Therefore, the passage is relevant because it provides information about CO2's role in limiting photosynthesis, which is part of the query. However, since it doesn't address light intensity, it might not fully answer the comparison. But the question is whether the passage is relevant, not if it's complete. Since the passage does discuss one of the factors (CO2) in the query, it is relevant. The answer should be true. | |
| $R$ | **StandardRanker**: 0.321
**Rank1**: 0.999
**Rank1-NoReason**: 0.998 | |

Table 20: An example of Rank1's (Qwen2.5-7B) reasoning process where it scores a partially relevant passage as highly relevant. As reference, we additionally provide the probability of relevance, $R$, for StandardRanker, Rank1, and Rank1-NoReason for the provided query-passage pair.

