# OpenReview forum: "Don’t "Overthink'" Pointwise Reranking: Is Reasoning Truly Necessary?"
_ICLR.cc/2026/Conference — Submitted to ICLR 2026_

### Official Review · Reviewer_JX9a · 2025-10-28

**Soundness:** 2
**Presentation:** 3
**Contribution:** 2
**Rating:** 4
**Confidence:** 4

**Summary:**

This paper focus on  investigating wether reasoning actually improve pointwise reranking accuracy by comparing reasoning-based pointwise rerankers (Rank1) to standard, non-reasoning pointwise rerankers (StandardRanker) under identical training conditions, and observe that StandardRanker generally outperforms Rank1. Building on this observation, This paper studys the importance of reasoning to Rank1 by disabling its reasoning process (Rank1-NoReason), and find that Rank1-NoReason is surprisingly more effective than Rank1. Experimental results reveal that pointwise reasoning rerankers are bottlenecked by the LLM's reasoning process, which pushes it toward polarized relevance scores and thus fails to consider the partial relevance of passages, a key factor for the accuracy of pointwise rerankers.

**Strengths:**

1. This paper fills this gap by providing a thorough comparison between reasoning-based rerankers and their equivalent non-reasoning counterparts under identical backbones and training data.
2. This paper is very easy to read and clearly conveys the authors’ intentions, making it easy to follow.
3. The experiments in this paper are highly rigorous, with all potential confounding factors carefully controlled to ensure the validity and reliability of the results. This paper demonstrate strong experimental discipline, conducting systematic analyses to isolate the effects of reasoning from other influencing variables.

**Weaknesses:**

1. In general, the paper focuses on aligning existing methods under comparable settings and reporting the resulting findings, without introducing new techniques or theoretical contributions. Hence, its novelty is rather limited.
2. In the implementation of Rank1-NoReason, the authors use <think> Okay, I have finished thinking. </think>. This implementation may not be entirely appropriate. I believe a more precise approach would be to use a blank (empty string) instead. According to existing studies [1][2], using logically invalid Chain-of-Thought (CoT) prompting can improve performance; therefore, merely including a <think> marker when removing reasoning is arguably not entirely justified.  [1] Invalid Logic, Equivalent Gains: The Bizarreness of Reasoning in Language Model Prompting [2] ThinkPilot: Steering Reasoning Models via Automated Think-prefixes Optimization
3. According to the assumptions in this paper, adding reasoning tends to polarize relevance scores, which reduces the performance of pointwise rerankers. Figure 2 shows that the score distribution of Rank1-NoReason is smoother compared to Rank1. However, despite its smoother distribution, Rank1-NoReason still does not achieve the same performance as StandardRanker.

**Questions:**

1. Can the authors provide a further explanation for the phenomenon observed in the third paragraph of Section 4.2?

---

> ### Author Response · Authors · 2025-11-24
>
> Thank you for your review! We are glad you found that our study fills an important gap and that experiments are highly rigorous. We hoped to ensure that we can truly understand why reasoning is not helpful.
>
> **In general, the paper focuses on aligning existing methods under comparable settings and reporting the resulting findings, without introducing new techniques or theoretical contributions. Hence, its novelty is rather limited.**
>
> > As we mention to reviewer d73v, we believe that our core conceptual novelty of our paper is the analysis that identifies why Rank1’s reasoning harms pointwise reranking effectiveness. To the best of our knowledge, no one has challenged the necessity of the reasoning process for LLMs in information retrieval tasks. In fact, most recent work in the reasoning for reranking literature (as we mention in the introduction) has explicitly assumed that it is helpful without any valid justification.
>
> > We believe our results are critical for designing future IR systems that are effective, whether it be for building more effective reasoning-based systems or encouraging others to focus on methods that don’t require a reasoning chain.
>
> **In the implementation of Rank1-NoReason, the authors use [object Object] Okay, I have finished thinking. [object Object]. This implementation may not be entirely appropriate. I believe a more precise approach would be to use a blank (empty string) instead. According to existing studies [1][2], using logically invalid Chain-of-Thought (CoT) prompting can improve performance; therefore, merely including a [object Object] marker when removing reasoning is arguably not entirely justified. [1] Invalid Logic, Equivalent Gains: The Bizarreness of Reasoning in Language Model Prompting [2] ThinkPilot: Steering Reasoning Models via Automated Think-prefixes Optimization**
>
> > Thank you for bringing this up! The reason we opted to do [object Object] Okay, I have finished thinking. [object Object] is mainly because Rank1 was trained to always see some text between its thinking tokens. So we aimed to mimic the case where the LLM only has access to its final reasoning sentence, i.e., “Okay, I am done thinking”, but without its reasoning substance (i.e., its thinking on the relevance of the passage to the query).
>
> > Your point is exactly what we aimed to study in Table 4, where we test with different reasoning chains. In our newly updated version of the paper, in Table 11 we further test on BRIGHT with the blank (empty string). We note that regardless, our findings remain: Removing reasoning is competitive or even better than Rank1 across most of MS MARCO and BRIGHT.
>
> **According to the assumptions in this paper, adding reasoning tends to polarize relevance scores, which reduces the performance of pointwise rerankers. Figure 2 shows that the score distribution of Rank1-NoReason is smoother compared to Rank1. However, despite its smoother distribution, Rank1-NoReason still does not achieve the same performance as StandardRanker.**
>
> > To clarify, we did not explicitly mean that smoother distributions automatically implies stronger performance. There are two factors that we believe matter: (1)  the ability to capture a relevance score which implicitly scores the most relevant passages higher than less relevant passages and (2) accuracy; a more accurate relevance classifier will have more effectiveness. As StandardRanker is more accurate at classification that Rank1-NoReason, it will perform better.
>
> > Hopefully this makes things clearer! We added a discussion with empirical evidence in Section 4.4 and Appendix D.
>
> **Can the authors provide a further explanation for the phenomenon observed in the third paragraph of Section 4.2?**
>
> > Yes! We believe the cause of this phenomenon is that when the Rank1 generates a reasoning process, computing Equation 2 becomes fairly one sided since the model made its prediction in its reasoning chain (as we show in Table 6). In its training, it is likely that whenever the reasoning chain makes its relevance conclusion, it was identical to the downstream label. So in turn Rank1 doesn’t learn any uncertainty: P(true) given the reasoning states true is always high and vice versa.

---

> > ### Author Response · Authors · 2025-11-27
> > **Feedback on our rebuttal**
> >
> > Thank you once again for your insightful review! Since the discussion period will end in a few days, we wanted to make sure that we have adequately addressed the issues you raised. We also hope that our additional experiments have addressed most of your concerns. We would appreciate your feedback on our responses!

---

### Official Review · Reviewer_d73v · 2025-10-30

**Soundness:** 3
**Presentation:** 3
**Contribution:** 2
**Rating:** 4
**Confidence:** 5

**Summary:**

This paper investigates whether explicit reasoning actually benefits LLM based pointwise rerankers. The experiments are inspired by Rank1, the authors conduct controlled experiments comparing three variants: 1) StandardRanker (direct relevance classification), 2) Rank1 (reasoning-before-prediction), 3) Rank1-NoReason (reasoning disabled). All using the same Qwen2.5 backbones, data, and LoRA fine-tuning.

Across MS MARCO and BRIGHT datasets, the paper finds that reasoning consistently reduces effectiveness: both StandardRanker and Rank1-NoReason outperform Rank1, with reasoning producing more polarized relevance scores and weaker modeling of partial relevance. The authors further test self-consistency and inverse-training variants but find limited improvement. The authors claim that reasoning, in its current form, is unnecessary or even harmful for pointwise reranking.

**Strengths:**

1. Well-controlled empirical setup.
The paper carefully eliminates confounding variables with same backbone, training data, and fine-tuning recipe, enabling a fair and reproducible comparison rarely seen in prior reasoning works.

2. Clear exposition and empirical transparency.
Figures and tables are easy to follow, prompts and code are documented, and reasoning examples (Table 6) illustrate the claimed polarization effect intuitively.

3. Timely contribution.
The work addresses a currently popular but under-examined assumption that reasoning always improves LLM performance, making its negative results relevant to both the IR and reasoning communities.

**Weaknesses:**

1. limited novelty. While the empirical setup is rigorous, the overall conceptual novelty is limited. The study primarily revisits the question, whether reasoning-before-prediction improves ranking performance, through replication and controlled comparison, but does not introduce new modeling, training, or theoretical innovations. The work therefore reads more as a careful diagnostic replication of Rank1 rather than a forward methodological contribution or a new hypothesis about how reasoning could be made effective.

2. Limited exploration of alternative reasoning placements or purposes. The study focuses exclusively on the “reasoning-before-prediction” paradigm, leaving unexplored other plausible reasoning designs that may capture complementary benefits, such as reasoning-after-prediction (for explainability) or reasoning-as-regularization (for robustness). Reasoning could hold value beyond accuracy, e.g., by improving model interpretability, calibration under distributional shifts, or user trust. Without examining whether reasoning contributes to robustness (e.g., under adversarial or out-of-domain perturbations) or to stable explanation quality, it remains unclear whether the observed degradation reflects a fundamental flaw of reasoning or an artifact of this specific evaluation setup.

3. Undertrained and potentially biased reasoning models. The negative finding that reasoning reduces performance may partly arise from the tuning strategy rather than an inherent weakness of reasoning itself. First, all models are fine-tuned with LoRA for only one epoch, using a fixed configuration (rank 32, α=64). Such shallow adaptation may under-optimize reasoning sequences, which typically require longer or multi-stage training to align reasoning tokens with downstream labels. Second, the reasoning data themselves, generated by DeepSeek-R1, may contain stylistic bias or overconfident language that systematically distorts supervision. The paper does not quantify how this bias propagates into Rank1’s learning. Without such analysis, it is difficult to determine whether reasoning failure stems from model capacity, data bias, or tuning procedure.

4. Lack of quantitative calibration analysis. The central claim that reasoning polarizes probabilities and harms modeling of partial relevance, is plausible but remains qualitatively supported. Figure 2 visually suggests this polarization, yet no quantitative evidence is presented to substantiate or measure its magnitude. Including calibration or uncertainty metrics such as entropy or KL divergence between score distributions would provide objective validation of this hypothesis. These measures could also reveal whether reasoning affects confidence miscalibration (i.e., overconfident “true” predictions) differently across datasets or model sizes. Without such analysis, the explanation for why reasoning fails remains speculative and descriptive rather than empirically grounded.

5. Limited interpretability and trade-off analysis. While the paper claims reasoning degrades accuracy, it does not explore whether reasoning offers compensatory benefits in interpretability or case-specific generalization. For example, reasoning might improve performance on complex or long queries, even if it hurts on simpler ones. Similarly, reasoning could help explain borderline relevance decisions even if it does not change ranking order. However, the study treats reasoning as monolithic, without differentiating by query difficulty, reasoning length, or reasoning structure. As a result, potential trade-offs between reasoning depth and ranking effectiveness are not analyzed, leaving open whether “turning off reasoning” truly yields a better system overall or simply optimizes for one narrow metric (nDCG@10) at the expense of transparency and task diversity.

**Questions:**

1. Your study convincingly shows that reasoning-before-prediction harms pointwise reranking accuracy. However, could you clarify whether your claim is intended to generalize beyond this paradigm? For instance, do you believe your conclusion extends to reasoning-after-prediction or reasoning-as-regularization frameworks? A clearer articulation of scope would help contextualize how general the “reasoning is unnecessary” statement should be interpreted.

2. The paper fine-tunes each model for only one epoch with fixed LoRA settings. Have you explored whether extending fine-tuning duration, adjusting LoRA ranks, or using multi-stage optimization would change the observed trend? In particular, could undertraining explain why reasoning appears ineffective at all model scales?

3. Since all reasoning chains originate from DeepSeek-R1, how do you assess or control for potential bias or stylistic artifacts in this synthetic data? Have you examined whether reasoning length, verbosity, or lexical patterns correlate with ranking degradation?
Quantifying how reasoning data quality affects downstream reranker performance would help separate data bias from model behavior.

4. The paper makes extensive use of long dash constructions (“—....—”), a stylistic pattern often characteristic of ChatGPT-generated writing. Could the authors clarify to what extent AI tools were used in manuscript drafting or editing? (i.e., line 50, line 61, line 62, line 189, line 208)

---

> ### Author Response · Authors · 2025-11-24
>
> Thank you for your comments and feedback! We are super glad you find our contribution both timely and relevant. That is our goal. We hope our findings can be helpful to the NLP and IR communities!
>
> **1. limited novelty...**
> > We believe that the core conceptual novelty of our paper is the analysis that identifies why  reasoning  harms pointwise reranking effectiveness. To the best of our knowledge, no one has challenged the necessity of the reasoning process for LLMs in information retrieval tasks (as you mention, this is under-examined!). In fact, most recent work in the reasoning for reranking literature (as we mention in the introduction) has explicitly assumed that it is helpful without any valid justification.
>
> > We believe our results are critical for designing future IR systems that are effective, whether it be for building more effective reasoning-based systems or encouraging others to focus on methods that don’t require a reasoning chain.
>
> **2. Limited exploration of alternative reasoning placements or purposes...**
>
> > This exploration was thoroughly explored and discussed in Section 5 and in Table 7. We did in fact find that reasoning after prediction is a viable option for those who want the potential explainability benefits. One benefit here is that by predicting before, you can get the benefits of both worlds, where the reasoning can be shared with the user if requested. (Though it is debatable if for search a user wants this; this would require future work outside the scope of our paper.)
>
> > Regarding robustness, I think our results strongly suggest that this is not the case. We tested extensively on 13 datasets, with 3 sized LLMs. Our results were clear: out-of-domain, reasoning was less effective than no reasoning.
>
> **3. Undertrained and potentially biased reasoning models....**
>
> > Regarding the tuning strategy, this was actually considered in our experimental setup. Our experiments found no advantage to training for longer. The original Rank1 implementation that we used was tuned for multiple-epochs and checkpoints were  selected based on early-stopping (See [1] for details). Our Rank1 implementation, while only tuned for one epoch, was competitive with the original Rank1 (as shown in our results). Thus, it is unlikely that the reduction in performance is due to tuning strategy. Regardless of how long Rank1 was trained, StandardRanker performed better (even though, StandardRanker itself was not fully tuned).
>
> [1] Weller et al., Rank1: Test-Time Compute for Reranking in Information Retrieval
>
> **4. Lack of quantitative calibration analysis.**
>
> > We added a discussion in our new uploaded paper with empirical evidence of how score polarization harms nDCG (in Section 4.4) and how reasoning harms the modeling of partial relevance (in Appendix D).
>
> **5. Limited interpretability and trade-off analysis.**
>
> > Regarding generalization: This is why we evaluated on the BRIGHT benchmark, which captures more complex and longer queries compared to MS MARCO. Furthermore, the BRIGHT and MS MARCO datasets capture a wide variety of tasks: web search, mathematics, coding, etc. Our results clearly demonstrated that regardless of the task or query-style, reasoning harmed effectiveness. We believe our evaluation explores each of the cases you mention.
>
> > Furthermore, as mentioned above, we do consider the possible benefits of interpretation in our discussion where we propose a strategy which balances accuracy and interpretability by training to predict-then-reason. We note, however, further exploration of trade-offs would require a more comprehensive study with human-subjects, which is out-of-scope for our paper.

---

> > ### Author Response · Authors · 2025-11-24
> >
> > Response to Questions:
> >
> > **Q1. Your study convincingly shows that reasoning-before-prediction harms pointwise reranking accuracy....**
> >
> > > As we mention above, we explore each of the cases that are mentioned in our discussion. As we state in our paper, we believe our results support the following conclusion: in Rank1’s current state, reasoning is not necessary, or useful, for pointwise rerankers at inference time.
> >
> > **Q2. The paper fine-tunes each model for only one epoch with fixed LoRA settings....**
> >
> > > We answer this above: TLDR, our experiments find no benefits to additional fine-tuning for Rank1. Thus, we do not believe differences in effectiveness are due to undertraining. We believe it is due to the score polarization caused by Rank1's reasoning process.
> >
> > **Q3. Since all reasoning chains originate from DeepSeek-R1, how do you assess or control for potential bias or stylistic artifacts in this synthetic data?...**
> >
> > > Thanks for the suggestion. We have explored shortening the reasoning chains at inference time, but found no improvements by doing such. However, our paper aimed to focus on understanding how Rank1 performs in comparison to baselines under equal settings. Studying manipulations on the data, while interesting, is out of scope for our paper. However, we believe this is an interesting direction for future work: Now that we know why and how reasoning models can fail for reranking (the goal of our study) we can begin making improvements
> >
> > **Q4: The paper makes extensive use of long dash constructions (“—....—”), a stylistic pattern often characteristic of ChatGPT-generated writing.**
> >
> > > As we mention in Appendix A, LLMs were solely utilized to assist in making grammatical edits for sections of this work, including the text and tables.

---

> > > ### Author Response · Authors · 2025-11-27
> > > **Feedback on our rebuttal**
> > >
> > > Thank you once again for all your helpful feedback! With the discussion period ending in a few days, we wanted to ensure that we adequately addressed the issues you raised. We would appreciate your feedback on our responses!

---

### Official Review · Reviewer_KvpF · 2025-10-31

**Soundness:** 3
**Presentation:** 3
**Contribution:** 2
**Rating:** 6
**Confidence:** 4

**Summary:**

This paper investigates whether introducing reasoning processes, such as chain-of-thought (CoT), can genuinely improve the performance of large language model (LLM)-based pointwise reranking tasks. The authors compare a reasoning-enhanced reranker (Rank1) with a standard reranker without reasoning (StandardRanker) under identical model architecture (Qwen2.5), training data, and hyperparameters. The experiments reveal that StandardRanker consistently outperforms Rank1 on both in-domain (MS MARCO) and out-of-domain (BRIGHT) datasets. Interestingly, disabling the reasoning steps in Rank1 during inference (Rank1-NoReason) leads to further performance improvements.

**Strengths:**

1. The effectiveness of reasoning training for LLMs in ranking tasks addresses an important research question.
2. The paper identifies an intriguing phenomenon: CoT knowledge distillation in supervised fine-tuning (SFT) may not necessarily enhance model reasoning capabilities.
3. The manuscript is well-written with clear and coherent argumentation.

**Weaknesses:**

Lack of statistical significance testing: The paper does not verify the statistical significance of experimental results, leaving the reliability of the findings uncertain.

Limited generalizability: Experiments are conducted solely on the Qwen 2.5 model series. Whether the observed “reasoning harm” phenomenon extends to other mainstream LLMs (e.g., Qwen3, LLaMA series) remains unverified, thereby limiting the universality and persuasiveness of the conclusions.

Unexplored influence of label form: The use of true/false string labels during training could affect the results. It remains unclear whether alternative label formats, such as binary relevance scores (0/1) or yes/no tokens, would alter the conclusions.

Unquantified quality and impact of reasoning data (CoT):
The CoT used in training is generated by the Deepseek model, whose quality is difficult to assess. Whether higher-quality CoT generated by more advanced models (e.g., GPT-5 or Gemini 2.5 Pro) would yield different outcomes is unexplored.
Potential distributional differences between Deepseek’s reasoning patterns and the Qwen 2.5 model’s “thinking style” may impede effective learning of reasoning capabilities, but this aspect is not discussed.
Table 4 lacks statistical significance testing, raising concerns that observed metric fluctuations may stem from input variance rather than generalizable effects.
Single retrieval method: The study employs only one retrieval method per dataset. It is uncertain whether findings are affected by the quality and characteristics of the retrieval pool. For instance, whether similar conclusions hold when reranking documents retrieved via pure BM25 on BRIGHT requires verification.

Insufficient empirical support for the attribution of ‘score polarization’:

The paper attributes performance degradation to score polarization caused by reasoning, but this remains a logically plausible hypothesis rather than a rigorously validated conclusion.

Fundamental questions such as the cause of score polarization, zero-shot score distributions, and whether model training generally promotes polarized or balanced scores remain unaddressed.

Observations that Qwen3 achieves strong zero-shot performance on BRIGHT despite apparently polarized score distributions suggest a discrepancy with the paper’s reasoning.
More importantly, causal evidence linking score polarization directly to performance decline is lacking.

Missing related work section: The paper lacks a dedicated related work discussion and does not systematically review or compare with prominent reasoning-based ranking models such as RankR1, REARANKER, and TFRank.

**Questions:**

See Weaknesses

---

> ### Author Response · Authors · 2025-11-25
>
> Thank you for your feedback and comments!!! We are happy that you find our research to be addressing an important research question. We hope that our comments below and fixes to the paper can help clarify your concerns!
>
> **Lack of statistical significance testing: The paper does not verify the statistical significance of experimental results, leaving the reliability of the findings uncertain.**
>
> **Table 4 lacks statistical significance testing, raising concerns that observed metric fluctuations may stem from input variance rather than generalizable effects**
>
> > We have added statistical significance results to the main tables (Table 1 and 2) as well as Table 4 in our updated version of the paper!
>
> **Limited generalizability: Experiments are conducted solely on the Qwen 2.5 model series. Whether the observed “reasoning harm” phenomenon extends to other mainstream LLMs (e.g., Qwen3, LLaMA series) remains unverified, thereby limiting the universality and persuasiveness of the conclusions.**
>
> > Per your suggestion, we added results with Qwen3-4B for both StandardRanker and Rank1. These results can be found in Appendix K. Generally, we found a similar conclusion. StandardRanker outperforms Rank1 even with Qwen3. This further confirms the generalizability of our findings (StandardRanker achieved 61.5 average nDCG@10 on MS MARCO and Rank1,  56.8 average nDCG@10.
>
> **Unexplored influence of label form: The use of true/false string labels during training could affect the results. It remains unclear whether alternative label formats, such as binary relevance scores (0/1) or yes/no tokens, would alter the conclusions.**
>
> > Unfortunately, we opted against this experiment due (1) to compute availability and (2) limited space. To create different labels, it would require modification of the reasoning to be fair across different label forms. As shown in the Rank1 paper, the procedure to create the Rank1 data is quite extensive. Due to this, we opted on other experiments to better understand the cause of why reasoning causes pointwise rerankers to fail.
>
> > Furthermore,  it’s actually been shown that the label form makes a negligible impact on effectiveness of text generation-based rerankers when trained with an amount of data similar in scale to Rank1’s training data [2].
>
> [1] Rank1: Test-Time Compute for Reranking in Information Retrieval
> [2] Document Ranking with a Pretrained Sequence-to-Sequence Model
>
> **Unquantified quality and impact of reasoning data (CoT): The CoT used in training is generated by the Deepseek model, whose quality is difficult to assess. Whether higher-quality CoT generated by more advanced models (e.g., GPT-5 or Gemini 2.5 Pro) would yield different outcomes is unexplored. Potential distributional differences between Deepseek’s reasoning patterns and the Qwen 2.5 model’s “thinking style” may impede effective learning of reasoning capabilities, but this aspect is not discussed.**
>
> > We would like to note that at the time of our writing, DeepSeek was one of the few models that offered open-source access to the models reasoning process to enable distillation. In fact, to the best of our knowledge GPT-5 and Gemini 2.5/3 do not provide the full reasoning chain to prevent people from distillation. While it is possible that Rank1 may improve with better reasoning data, the exact process for generating such data is not trivial, as described in [1], thus doing so was out of scope for our paper. As such, we opted to focus on why DeepSeek’s reasoning chains harm pointwise rerankers. However, we believe that our insights generalize regardless of the model: it is critical that the reasoning chains do not push the reranker towards polarized scores. We believe this is a great direction for future work, thanks for bringing it up!
>
> **Single retrieval method: The study employs only one retrieval method per dataset. It is uncertain whether findings are affected by the quality and characteristics of the retrieval pool. For instance, whether similar conclusions hold when reranking documents retrieved via pure BM25 on BRIGHT requires verification.**
>
> >We have some results for this on BRIGHT! We reranked the ReasonIR first-stage results and got the following scores:
>
>
> |                   | NDCG@10 |
> |-------------------------------|---------|
> | **BM25 + GPT-4 CoT**          | 27.0    |
> | + StandardRanker (7B)         | 31.0    |
> | + Rank1 (7B, Our Impl.)       | 27.8    |
> | **ReasonIR + GPT-4 CoT**      | 30.5    |
> | + StandardRanker (7B)         | 31.2    |
> | + Rank1 (7B, Our Impl.)       | 25.8    |
>
>
> >Interestingly, StandardRanker remains stable in terms of nDCG@10, whereas Rank1 does worse! Our results seem to show consistency with our findings regardless of the first stage retriever. We have included these results in Section J of our paper.

---

> > ### Author Response · Authors · 2025-11-25
> >
> > **Insufficient empirical support for the attribution of ‘score polarization’: The paper attributes performance degradation to score polarization caused by reasoning, but this remains a logically plausible hypothesis rather than a rigorously validated conclusion.**
> >
> > **Fundamental questions such as the cause of score polarization, zero-shot score distributions, and whether model training generally promotes polarized or balanced scores remain unaddressed.**
> >
> > **More importantly, causal evidence linking score polarization directly to performance decline is lacking.****
> >
> > > Thank you for mentioning this. We have added a deeper investigation of score polarization in Section 4.4 and Appendix D.
> >
> > > TLDR: While not casual, we have shown correlationally that the ability to better capture the relative relevance between pairs of passages has a significant and positive effect on nDCG, meaning that yes, we can directly tie capturing partial relevance (i.e., not assigning the same score to all passage) to higher effectiveness of rerankers with regards to nDCG!
> >
> >
> > **Observations that Qwen3 achieves strong zero-shot performance on BRIGHT despite apparently polarized score distributions suggest a discrepancy with the paper’s reasoning.**
> >
> > > Could you clarify which Qwen3 you are referring to? Is it Qwen3-Reranker? Yes, that model is very strong! But actually, for that model they “turn off” thinking as well. (I.e., the model does not perform any reasoning prior to making a relevance prediction.) See the prompt at the end of Page 3 here in [3] , they do “< think >\n\n< /think >\n\n”.
> >
> > [3] Qwen3 Embedding: Advancing Text Embedding and Reranking Through Foundation Models
> >
> > **Missing related work section: The paper lacks a dedicated related work discussion and does not systematically review or compare with prominent reasoning-based ranking models such as RankR1, REARANKER, and TFRank.**
> >
> > > Our Section 5 actually is a related work section! We overview the literature that has questioned the CoT process,  and also discuss many baselines in the reasoning reranking space, including Rank-R1 and other state-of-the-art reasoning models in the IR literature. Our aim was to loop it in with empirical evidence (via our experiments) to help guide our discussion in the context of our findings.
> >
> > > Additionally, Table 8 does in fact compare against prominent reasoning-based ranking models. We selected Rank-R1, Rank-K, and ReasonRank since they were the highest performing baselines at the time of our writing.

---

> > > ### Author Response · Authors · 2025-11-27
> > > **Feedback on our rebuttal**
> > >
> > > Thank you once again for your helpful review! Since the discussion period ends in a few days, we wanted to ensure that we have adequately addressed the issues you raised. We also hope our additional experiments have addressed most of your concerns. We would appreciate any feedback on our responses!

---

> > > > ### Comment · Reviewer_KvpF · 2025-11-28
> > > >
> > > > Thank you to the authors for their response. I have no further questions.

---

### Official Review · Reviewer_ohBn · 2025-11-01

**Soundness:** 3
**Presentation:** 3
**Contribution:** 2
**Rating:** 2
**Confidence:** 4

**Summary:**

The authors of this paper investigate whether reasoning is beneficial for point-wise reranking performance. They demonstrate that Rank1, a recently proposed reasoning based reranker, performs better when forced to not produce any reasoning tokens. Additionally, the authors hypothesize that reasoning hurts performance because it results in a bimodal probability distribution for the final yes/no token that is used to compute the reranker score.

**Strengths:**

The strengths of the paper are:
- The authors correctly point out that the the primary non-reasoning baselines used in Rank1 are built on older language models and therefore the baseline comparison is flawed in the Rank1 paper.
- The proposed model, StandardRanker, outperforms Rank1 without using reasoning traces.
- The analysis done with Rank1-NoReason, that is the Rank1 model with reasoning disables, is interesting.

**Weaknesses:**

The weaknesses of the paper are:
- The authors claim that the Rank1 has worse performance because the partial relevance of query-passage pairs is not modeled by Rank1. I agree that the relevance scores from Rank1 are either 0 or 1 most of the time. But I'm not convinced that this is the primary reason for why Rank1 underperforms. In the datasets that the authors use (MSMarco and Bright), the samples are either relevant or non-relevant (and there are no partially relevant samples). So it is important to model partial relevance? It is possible that modeling partial relevance provides a better training signal, but the authors have not clearly demonstrated the causal link between modeling partial relevance and improved performance.
- In appendix H, Rank1 is trained to produce non-binary relevance labels. The relevance scores from Rank1-hybrid should now be distributed in a less bimodal way. Can we see the distribution of scores from this model? How come this model still underperforms StandardRanker+hybrid?
- No analysis of whether reasoning helps listwise rerankers. The current scope of the paper is very limited and the analysis is mostly limited to RankR1.
- Shao et al. (2025) propose using Qwen models directly without training. StandardRanker is not compared to this baseline.

Reference:
Rulin Shao, Rui Qiao, Varsha Kishore, Niklas Muennighoff, Xi Victoria Lin, Daniela Rus, Bryan Kian Hsiang Low, Sewon Min, Wen-tau Yih, Pang Wei Koh, et al. ReasonIR: Training Retrievers for Reasoning Tasks. arXiv preprint arXiv:2504.20595, 2025.

**Questions:**

Here are some questions I have:
- Which dataset is used in figure 2?
- Which model and dataset are used for the analysis in section 5?

---

> ### Author Response · Authors · 2025-11-25
>
> Thank you for your feedback and comments!!! We are happy you found our results interesting. Below we respond to your comments and have added fixes in our newest version of the paper. Hopefully this can help clarify your concerns.
>
> **The authors claim that the Rank1 has worse performance because the partial relevance of query-passage pairs is not modeled by Rank1. I agree that the relevance scores from Rank1 are either 0 or 1 most of the time. But I'm not convinced that this is the primary reason for why Rank1 underperforms. In the datasets that the authors use (MSMarco and Bright), the samples are either relevant or non-relevant (and there are no partially relevant samples). So it is important to model partial relevance? It is possible that modeling partial relevance provides a better training signal, but the authors have not clearly demonstrated the causal link between modeling partial relevance and improved performance.**
>
> > Yes! We have included some additional details in Appendix D (text in blue) for why this is important. We wanted to also note that MS MARCO (and, specifically, DL19-DL23) actually use graded relevance: Perfectly Relevant > Highly Relevant > Related > Irrelevant.
>
> > TLDR: Rank1’s inability to produce relevance scores that can implicitly differentiate between the different nuances of relevance (e.g., Perfectly Relevant versus  Highly Relevant) cause it to be more negatively impacted when it makes a misclassification since its probabilities don’t implicitly produce lower scores for “less relevant” documents.  It always produces the same relevance score when it predicts “true” even if the document is Perfectly Relevant or, say, Related. On the other hand when StandardRanker and Rank1-NoReason score a document as "true", they naturally score Perfectly Relevant passages higher than Highly Relevant or Related passages,  on average. This is critical for a high nDCG score.
>
> > Furthermore, while not casual, in section 4.4, we have shown correlationally that the ability to capture the relative relevance between pairs of passages has a significant positive effect on nDCG, meaning that yes, capturing partial relevance (i.e., not assigning the same score to all passage) is important to performance of rerankers with regards to nDCG.
>
> > Hopefully this clarifies!
>
>
> **In appendix H, Rank1 is trained to produce non-binary relevance labels. The relevance scores from Rank1-hybrid should now be distributed in a less bimodal way. Can we see the distribution of scores from this model? How come this model still underperforms StandardRanker+hybrid?**
>
> > We added this plot to Appendix I (this was the previous Appendix H)! We hypothesize this model underperforms StandardRanker+hybrid simply because StandardRanker is a stronger ranker than Rank1 (non-binary labels). Both get the benefit of the hybrid scores, but likely the original rankings from StandardRanker are stronger, thus making it more effective.
>
> **No analysis of whether reasoning helps listwise rerankers. The current scope of the paper is very limited and the analysis is mostly limited to RankR1.**
>
> > We believe, given that pointwise rerankers are still very popular due to their efficiency, they warranted their own in-depth and dedicated analysis.
>
>
> **Shao et al. (2025) propose using Qwen models directly without training. StandardRanker is not compared to this baseline.**
>
> > We did not include zero-shot models as that would be out of scope of our research question. The aim of our study was to directly compare reasoning and non-reasoning pointwise rerankers under controlled test: both trained on the same data with the same backbone base LLM. However, we have included this baseline in Table 13 as reference as it fits the Hybrid setup of the experiments discussed in Appendix H.
>
> > We would like to note that the QwenReranker from Shao et al., further confirms our point: Simple, non-reasoning, rerankers are stronger than a reasoning reranker.
>
>
> **Which dataset is used in Figure 2? Which model and dataset are used for the analysis in Section 5?**
>
> > Figure 2 and Section 5 are based on rerankers trained using Qwen2.5-7B on the DL19 dataset . We have updated the paper to make this more clear.

---

> > ### Author Response · Authors · 2025-11-27
> > **Feedback on our rebuttal**
> >
> > Thank you once again for your insightful review! Since the discussion period will end in a few days, we wanted to make sure that we have adequately addressed the issues you raised. We also hope that our responses and additional experiments/results have addressed most of your concerns. We would appreciate your feedback on our responses!

---

### Author Response · Authors · 2025-11-26
**General response to all the reviewers**

We appreciate the constructive and insightful comments from all the reviewers! We have provided detailed answers to the comments and questions from each reviewer in the different author responses. We also made the following updates to the PDF file based on the feedback from reviewers, with all changes highlighted in blue.

| Content                                             | Section              | Based on the comments from |
|-----------------------------------------------------|----------------------|-----------------------------|
|  Quantitatively demonstrating link between modeling partial relevance and improved performance     | Section 4.4, Appendix D          |  Reviewer ohBn, KvpF, d73v, JX9a          |
|  Added plot of Hybrid, Rank1 (non-binary labels) distribution     | Appendix I           | Reviewer ohBn              |
|  Comparison of StandardRanker to Shao et al. (2025) Qwen Reranker     |  Table 14          | Reviewer ohBn              |
| Added more details to Figures and analysis | Figure 2, Table 7 | Reviewer ohBn              |
| Statistical significance tests             | Table 1, Table 2, Table 4 |   Reviewer KvpF            |
| Added experiments on generalizability             | Appendix K |     Reviewer KvpF         |
| Added experiments on different retrieval methods             | Appendix J |     Reviewer KvpF         |
| Added more results with blank (empty string) reason for Rank1-NoReason            | Table 11 |   Reviewer JX9a        |

**General summary:**

A concern each of the reviewers mentioned is a better explanation for why partial reasoning harms reranker effectiveness alongside some quantitative evidence that backs this. This was our new addition to the paper in sections 4.4 and Appendix D. We believe this provides further evidence of our conclusions and claims.
Additionally, we  addressed the concerns reviewers had on trustworthiness of our results. We have added significance tests where available, experiments with different retrievers as well as different LLMs. We also included further results of Rank1-NoReason with an empty string reasoning process. Lastly, we made additional edits to our paper to make our tables and results more clear.


Reviewers also mentioned the following weaknesses, which we believe our paper already has addressed:

* **Exploration of alternative reasoning placements**: This is explored in Section 5, paragraph 3.

* **Missing related work section**: Our paper has an explicit related work section that is mixed with our discussion

* **Comparison to prominent reasoning-based rerankers**: Table 8 and our last paragraph in Section 5 discuss this

* **Undertrained models**: This was implicitly considered in our experiments when we use the original Rank1 checkpoints; These checkpoints were trained for multiple epochs and were selected using early stopping. As we mention in line 204: "This validates that any differences in effectiveness between StandardRanker and Rank1 cannot be attributed to potential differences in training conditions"

* **Case-specific generalization and interpretability**: This was considered with our use of the MS MARCO and BRIGHT benchmarks which considers a diverse range of complex and different sets of queries. So we believe we do in fact differentiate based on query difficulty. We additionally do discuss the interpretability advantages in in our discussion (Section 5) and consider a version of Rank1 (Rank1 Inverse Training) which balances accuracy and interpretability

We did our best to address all of the possible weaknesses addressed in the reviewers comments that was missing in our paper.  We believe each of these new additions further validate the claims in our paper and make the conclusions stronger. We thank our reviewers for their feedback!

---

### Meta-Review · Area_Chair_M4JL · 2025-12-10

**Summary:**

This paper investigate if reasoning is useful for pointwise reranking. The authors investigate this problem exclusively by doing various experiment with Rank1, which is a reasoning based reranking method. From experiments the authors show that in most cases standard reranking model (i.e. no reasoning rerank) performs better as compared to rank1 based reranking.

The main concern is that the authors claims reasoning does not help pointwise reranking, however, the experiment only looks at rank1 and finetuning with Lora adpators. While rank1 is one way of using reasoning for pointwise reranking, there are various other ways of using reasoning to do pointwise reranking.

**Reviewer Concerns:**

See summary

**Reviewer Scores:**

ohBn 2
KvpF 6
d73v 4
JX9a 4

---

### Decision · Program_Chairs · 2026-01-26

Reject